# NeuroCycle: Physiologically Constrained Cycling for Generating Neural Information-Rich fMRI from EEG

## Abstract

Electroencephalography (EEG) provides millisecond-level temporal resolution but suffers from poor spatial precision, whereas functional magnetic resonance imaging (fMRI) offers fine-grained spatial detail at the expense of cost and latency. Leveraging their complementarity, an emerging direction is to synthesize fMRI from EEG, enriching EEG with spatial information while retaining its efficiency. However, existing EEG to fMRI generation methods often lack designs to preserve information completeness and neglect neurophysiological priors, leading to reconstructions that may appear plausible but fail to ensure neuroscientific validity. We introduce NeuroCycle, a cyclic EEG–fMRI generation framework that enforces information completeness and neuroscientific plausibility. It incorporates two neurophysiological priors: (i) a Cross-Modal ROI-wise Structural Module that aligns fMRI embeddings with EEG-derived correlation patterns to preserve regional organization, and (ii) an R2E Physiological Connectivity Guidance Module that supervises covariance matrices via Riemannian-to-Euclidean mapping to maintain functional connectivity. The bidirectional cycle (EEG→fMRI→EEG) further enforces information completeness and cross-modal alignment, ensuring that synthesized fMRI retains key neural information. Experiments on NODDI and Oddball datasets show consistent improvements over state-of-the-art baselines, producing sharper voxel-wise fMRI with richer neural information, preserved connectivity, and stronger cross-modal alignment.

## 1 Introduction

Electroencephalography (EEG) is one of the most widely used modalities for non-invasive brain signal acquisition due to its high temporal resolution, low cost, and accessibility Gevins et al. (1995). It has been extensively applied in clinical monitoring and neuromodulation. However, EEG suffers from inherently limited spatial precision, which restricts its effectiveness in tasks requiring fine-grained localization, such as precise diagnosis and prediction of neurological disorders Srinivasan (1999). In contrast, functional magnetic resonance imaging (fMRI) provides stable observations of brain states with high spatial resolution, making it particularly powerful for brain region localization and disease prediction. Yet, the acquisition of fMRI is expensive, relies on bulky equipment, and is constrained by low temporal resolution, preventing its deployment in real-time monitoring and large-scale clinical scenarios Menon & Goodyear (2001). In this context, EEG and fMRI provide complementary advantages across temporal and spatial dimensions Huster et al. (2012); Mulert et al. (2004). Both originate from the same underlying neural activity: EEG captures neuronal electrical discharges, while fMRI reflects the hemodynamic response. Based on this complementarity, an important research question naturally arises: Can we synthesize spatially fine-grained fMRI representations solely from EEG data, thereby retaining the efficiency and affordability of EEG while overcoming its spatial limitations?

This question has attracted increasing attention, and several pioneering studies follow a common pipeline that encodes EEG signals into embeddings and subsequently generates fMRI representations through transformer- or GAN-based frameworks (e.g., NT-ViT Lanzino et al. (2024), E2fGAN Roos et al. (2025)). However, existing methods face critical limitations (Figure 1(a)). First, existing methods generally lack explicit modeling and supervision of neurophysiological pri-

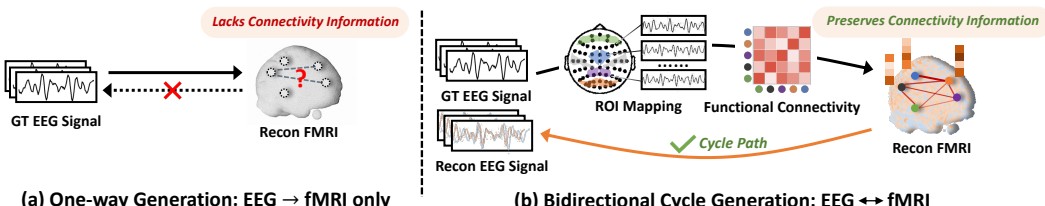

Figure 1: Traditional one-way EEG→fMRI generation (a) overlooks physiological priors and loses information completeness, whereas our cyclic framework (b) integrates connectivity priors and cycle consistency to preserve neural information and enhance biological plausibility.

ors, such as brain ROI-level organization and functional connectivity, making it difficult to ensure that the generated fMRI representations preserve complete and physiologically consistent information aligned with EEG data, thereby limiting their neuroscientific interpretability. Second, most approaches adopt a unidirectional EEG → fMRI mapping, neglecting the preservation of complete neurophysiological information. As a result, the synthesized fMRI may appear visually plausible but fail to guarantee that it faithfully retains the essential statistical characteristics embedded in EEG signals, which undermines its clinical utility and downstream applicability. Moreover, current methods fall short in capturing cross-modal consistency, limiting their ability to uncover and exploit the deep physiological coupling between EEG and fMRI.

To address these challenges, we propose a cyclic cross-modal EEG–fMRI generation framework (Figure 1(b)) that integrates neurophysiological priors to ensure both information completeness and neuroscientific plausibility. First, we incorporate neurophysiological priors through two dedicated modules: the Cross-Modal ROI-wise Structural Module, which aligns EEG-derived correlation patterns with reconstructed fMRI ROI embeddings to preserve regional organization, and the R2E Physiological Connectivity Guidance Module, which supervises EEG covariance matrices via Riemannian-to-Euclidean mapping to maintain functional connectivity consistency. Second, unlike unidirectional EEG → fMRI mappings that risk losing essential neural information, we introduce a pathway that regenerates EEG from the synthesized fMRI, thereby forming a cyclic bidirectional framework (EEG → fMRI → EEG) based on flow matching to guarantee that essential neural information is fully preserved. This cyclic constraint strengthens semantic alignment between modalities, ensuring that the generated fMRI captures both plausible spatial patterns and key neural signal statistics. Together, these innovations provide a principled framework for cross-modal brain signal generation, enhancing both neuroscientific interpretability and potential clinical utility.

In summary, our main contributions are threefold: (1) We propose the first cycle-based EEG–fMRI bidirectional generation framework via Flow Matching, which preserves the full and reversible neurophysiological details embedded in fMRI, thereby ensuring information completeness and modality alignment. (2) We incorporate neurophysiological priors through two modules: A Cross-Modal ROI-wise Structural Module that enforces consistency between EEG-derived connectivity patterns and reconstructed fMRI ROI embeddings, preserving brain regional organization; An R2E Physiological Connectivity Guidance Module that aligns EEG-derived and reconstructed EEG covariance matrices by mapping them from the Riemannian manifold to the Euclidean tangent space, enhancing functional connectivity interpretability and indirectly improving the physiological information integrity preserved in the reconstructed fMRI. (3) We validate the framework on NODDI Deligianni et al. (2016; 2014) and Oddball Walz et al. (2013; 2014; 2015) datasets, achieving superior fMRI reconstruction performance in preserving neural information and cross-modal consistency.

## 2 RELATED WORK

Early EEG-to-fMRI synthesis studies applied convolutional, graph-attentional, transformer, and adversarial designs to directly generate voxel-wise fMRI. CNN-TC Liu & Sajda (2019) used CNN-based transcoding but struggled with long-range dependencies. CNN-TAG Calhas & Henriques (2022) introduced graph attention over electrodes. NT-ViT Lanzino et al. (2024) employed self-attention for temporal–spectral relations. E2fGAN/E2fNet Roos et al. (2025) enhanced robustness under noise and shifts. Despite architectural diversity, these approaches are largely data-driven map-

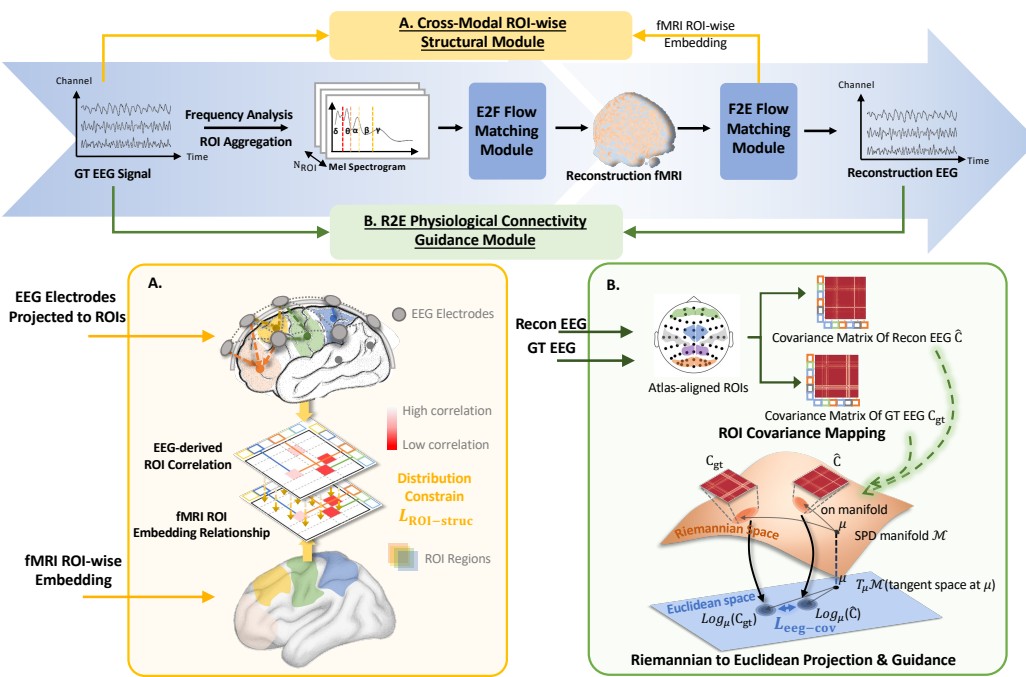

Figure 2: Overview of the proposed cyclic EEG–fMRI generation framework. EEG signals are mapped to fMRI via Flow Matching and back-translated to EEG for cycle consistency. Two guidance modules enhance neurophysiological plausibility: (A) Cross-Modal ROI-wise Structural Module for preserving ROI-level organization across modalities; (B) R2E Physiological Connectivity Guidance for functional connectivity alignment.

pings that do not enforce information integrity or neurophysiological constraints (e.g., connectivity priors), which may limit information completeness and biological plausibility in the synthesized volumes.

Distinct from the above voxel-wise setting, **NeuroBOLT** Li et al. (2024) performs ROI-level time-series synthesis in resting state via multi-dimensional feature mapping that fuses spatiotemporal and multi-scale spectral embeddings. While effective for parcellation-based dynamics and network analysis, its outputs are parcellation signals rather than 3D volumes, thereby discarding intra-ROI heterogeneity and precluding fine-grained spatial localization. In contrast, our **NeuroCycle** introduces a cyclic pathway (EEG → fMRI → EEG) to guarantee information completeness, while integrating neurophysiological priors to produce biologically constrained voxel-wise reconstructions that move beyond coarse ROI dynamics reconstruction.

## 3 METHOD

### 3.1 OVERVIEW

The overall framework of the EEG–fMRI Cyclic Generation Model is illustrated in Figure 2. For the EEG modality, the data of a subject with $C$ electrodes is denoted as $X^{\text{EEG}} = (x_1, x_2, \ldots, x_C) \in \mathbb{R}^{C \times T}$, where $x_i \in \mathbb{R}^T$ represents the temporal signal of the $i$-th electrode and $T$ is the length of the time window. For the fMRI modality, the volumetric data at a single time point is represented as $X^{fMRI} \in \mathbb{R}^{V_x \times V_y \times V_z}$, where $V_x, V_y, V_z$ denote the voxel dimensions along the three spatial axes. Our objective is to reconstruct fMRI signals $\hat{X}^{fMRI}$ from EEG inputs $X^{EEG}$ that not only appear realistic, but also faithfully preserve physiological structures and contain complete neural information compared to ground-truth fMRI $X^{fMRI}$. To achieve this, we design a cyclic mapping of EEG → fMRI → EEG. In this cycle, the reconstructed EEG $\hat{X}^{EEG}$ is compared against the ground-truth EEG $X^{EEG}$ to provide additional supervision. This cyclic constraint ensures that the

generated fMRI retains sufficient information to regenerate the original EEG, thereby guaranteeing both the information integrity of the reconstructed fMRI and the cross-modal consistency between EEG and fMRI.

To extract spectral information, the input raw EEG signal $X^{EEG}$ is transformed into a Mel-spectrogram representation, following the same preprocessing strategy adopted in Lanzino et al. (2024). This produces $X^{EEG}_{spec} \in \mathbb{R}^{C \times F \times T}$, with $F$ denoting the number of Mel frequency bands. Subsequently, electrodes are projected to atlas-defined cortical regions, and ROI-level spectral features are aggregated as $X^{EEG}_{roi} \in \mathbb{R}^{N_{roi} \times F \times T}$. We then input $X^{EEG}_{roi}$ into the EEG ROI encoder $E_{EEG}$ to obtain ROI-level latent embeddings $Z^{EEG}_{roi} \in \mathbb{R}^{N_{roi} \times d}$, where $d$ is the embedding dimension. These ROI embeddings are concatenated and projected through a projection layer to derive a compact global representation $Z^{EEG}_g$. The global embedding $Z^{EEG}_g$ is then fed into the EEG-to-fMRI (E2F) flow matching module $F_{E \to F}$ to predict the corresponding fMRI global embedding $Z^{fMRI}_g$. This embedding is further decoded by decoder $D_{fMRI}$ to reconstruct the volumetric fMRI data $\hat{X}^{fMRI}$. Next, the reconstructed fMRI $\hat{X}^{fMRI}$ is encoded by the fMRI encoder $E_{fMRI}$ into its ROI-level embedding $\hat{Z}^{fMRI}_{roi} \in \mathbb{R}^{N_{roi} \times d}$. This embedding is then fed into the proposed **Cross-Modal ROI-wise Structural Module**, which enforces that the fMRI ROI embeddings preserve the connectivity relationship inferred from EEG, thereby maintaining functional connectivity patterns that reflect structural organization and ensuring consistency across modalities. Then, following the procedure described above, a compact global representation obtained $\hat{Z}^{fMRI}_g$ is mapped to the EEG embedding space through the fMRI-to-EEG (F2E) flow matching module $F_{F \to E}$. The resulting representation is then passed into the EEG decoder $D_{EEG}$ to generate the reconstructed raw EEG signal $\hat{X}^{EEG} \in \mathbb{R}^{C \times T}$. The reconstructed EEG $\hat{X}^{EEG}$ together with the ground-truth EEG $X^{EEG}$ are fed into the proposed **R2E Physiological Connectivity Guidance Module**, which maps EEG-derived ROI-level functional connectivity matrices from the Riemannian manifold to the Euclidean tangent space and enforces agreement between the reconstructed and ground-truth EEG connectivity by minimizing their tangent-space discrepancy, thereby ensuring that the generated EEG preserves physiological connectivity relationships and, in turn, enhancing the information integrity of the generated fMRI.

## 3.2 ROI ENCODER AND BIDIRECTIONAL CROSS-MODAL FLOW MATCHING MODULE

To bridge the heterogeneous representations of EEG and fMRI, we design ROI-based encoders that project both modalities into a shared latent space while preserving global and regional brain dynamics, followed by bidirectional flow matching modules learning consistent cross-modal transformations. The module design is illustrated in Figure 3.

The raw EEG signal $X^{EEG} \in \mathbb{R}^{C \times T}$, is first transformed into a Mel-spectrogram representation $X^{EEG}_{spec} \in \mathbb{R}^{C \times F \times T}$, to incorporate spectral information. To aggregate electrode-level features into anatomically meaningful regions, we apply a predefined electrode-to-ROI mapping $\mathcal{M}_{EEG \to ROI}$, where each electrode contributes to a cortical ROI according to a distance-based weighting between the electrode location and the ROI center in the atlas Babiloni et al. (2004). Let $p_c \in \mathbb{R}^3$ denote the spatial coordinate of the $c$-th electrode, and $q_r \in \mathbb{R}^3$ denote the center coordinate of the $r$-th ROI. The distance between electrode $c$ and ROI $r$ is $d_{c,r} = \|p_c - q_r\|_2$. We define the unnormalized weight as the inverse distance $w_{c,r} = \frac{1}{d_{c,r} + \varepsilon}$, where $\varepsilon$ is a small constant to avoid division by zero. After normalization across all electrodes, the final weight is $\alpha_{c,r} = \frac{w_{c,r}}{\sum_{c'=1}^{C} w_{c',r}}$. Given the Mel-spectrogram representation of EEG signals $X^{EEG}_{spec}$, the ROI-level EEG spectral features for ROI $r$ are obtained as a weighted combination of electrode features: $X^{EEG}_{ROI}[r,f] = \sum_{c=1}^{C} \alpha_{c,r} \frac{1}{T} \sum_{t=1}^{T} X^{EEG}_{spec}[c,f,t]$, where $f = 1, \dots, F$ indexes Mel frequency bands.

These sequences, together with fMRI ROI representations, are encoded by modality-specific VAE-style encoders. Each encoder $E_{EEG}$ and $E_{fMRI}$ outputs ROI-level Gaussian parameters: $(\mu^{EEG}_{roi}, \log \sigma^{2,EEG}_{roi})$, $(\mu^{fMRI}_{roi}, \log \sigma^{2,fMRI}_{roi})$. ROI-level latent variables are sampled via the reparameterization trick: $z^m_{roi} = \mu^m_{roi} + \sigma^m_{roi} \odot \epsilon$, $\epsilon \sim \mathcal{N}(0, I)$, where $m \in \{EEG, fMRI\}$ denotes the modality.

To obtain a compact whole-brain representation, all ROI embeddings are first concatenated and then projected through a transformation function $f_{proj}$. This projected feature vector parameterizes

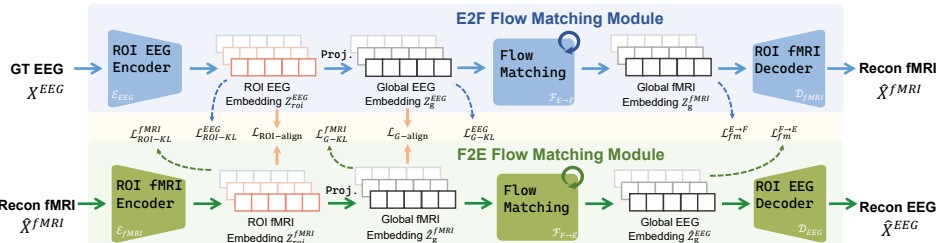

Figure 3: Illustration of the ROI Encoder and Bidirectional E2F/F2E Flowing Matching Module

a global Gaussian distribution: $(\mu_g^m, \log \sigma_g^{2,m}) = f_{\text{proj}}(\text{Concat}(z_{\text{roi}}^m))$, from which the global latent variable is sampled via the reparameterization trick: $z_g^m = \mu_g^m + \sigma_g^m \odot \epsilon, \quad \epsilon \sim \mathcal{N}(0, I)$, where $m \in \{\text{EEG}, \text{fMRI}\}$ denotes the modality.

On top of these latent encodings, two flow matching modules are introduced: E2F flow matching module $F_{E \to F}$ maps EEG latent variables into the fMRI space, while F2E flow matching module $F_{F \to E}$ maps fMRI latents into the EEG space. Given a source latent $z_s$ and a target latent $z_t$, the flow predicts $\hat{z}_t = F(z_s)$ and is trained with the mean-squared error loss $\mathcal{L}_{\text{fm}}(z_s, z_t) = |F(z_s) - z_t|^2$. The bidirectional flow losses are:

$$\mathcal{L}_{\text{fm}}^{E \to F} = \|F_{E \to F}(z_g^{\text{EEG}}) - z_g^{\text{fMRI}}\|_2^2, \quad \mathcal{L}_{\text{fm}}^{F \to E} = \|F_{F \to E}(z_g^{\text{fMRI}}) - z_g^{\text{EEG}}\|_2^2. \quad (1)$$

To regularize the latent space and stabilize training, KL divergence regularization is applied to both global and ROI-level distributions:

$$\mathcal{L}_{\text{KL}}^m = \mathcal{L}_{\text{G-KL}}^m + \mathcal{L}_{\text{ROI-KL}}^m = D_{\text{KL}}\big(q(z_g^m \mid X^m) \| \mathcal{N}(0, I)\big) + D_{\text{KL}}\big(q(z_{\text{roi}}^m \mid X^m) \| \mathcal{N}(0, I)\big). \quad (2)$$

The overall KL loss is then: $\mathcal{L}_{\text{KL}} = \mathcal{L}_{\text{KL}}^{\text{EEG}} + \mathcal{L}_{\text{KL}}^{\text{fMRI}}$.

Moreover, to explicitly align regional representations across modalities, we introduce an ROI-level correspondence loss $\mathcal{L}_{\text{ROI-align}}$, which enforces regional consistency between EEG and fMRI ROI embeddings. This loss minimizes the distance between paired EEG–fMRI ROI embeddings while contrasting against embeddings from other ROIs based on a contrastive InfoNCE objective. Similarly, global alignment between EEG and fMRI pairs latents is also encouraged. The two losses are defined as:

$$\mathcal{L}_{\text{ROI-align}} = -\frac{1}{N_{\text{ROI}}} \sum_{r=1}^{N_{\text{ROI}}} \log \frac{\exp(\text{sim}(z_{\text{roi},r}^{\text{EEG}}, z_{\text{roi},r}^{\text{fMRI}})/\tau)}{\sum_{r'} \exp(\text{sim}(z_{\text{roi},r}^{\text{EEG}}, z_{\text{roi},r'}^{\text{fMRI}})/\tau)}, \quad \mathcal{L}_{\text{align}} = -\log \frac{\exp(\text{sim}(z_g^{\text{EEG}}, z_g^{\text{fMRI}})/\tau)}{\sum_{z'} \exp(\text{sim}(z_g^{\text{EEG}}, z')/\tau)}. \quad (3)$$

where $z_{\text{roi},r}^{\text{EEG}}$ and $z_{\text{roi},r}^{\text{fMRI}}$ denote the $d$-dimensional embeddings of ROI $r$, $\text{sim}(\cdot, \cdot)$ is the cosine similarity, and $\tau$ is a temperature parameter.

Finally, the modality-specific decoders take the EEG and fMRI latents as input to reconstruct the original signals, $\hat{X}^{\text{fMRI}} = D_{\text{fMRI}}(F_{E \to F}(z_g^{\text{EEG}})), \quad \hat{X}^{\text{EEG}} = D_{\text{EEG}}(F_{F \to E}(\hat{z}_g^{\text{fMRI}}))$, with reconstruction losses is defined as the mean squared error (MSE) between the predicted and ground-truth signals:

$$\mathcal{L}_{\text{recon}}^m = \frac{1}{|X^m|} \|\hat{X}^m - X^m\|_2^2, \quad m \in \{\text{EEG}, \text{fMRI}\}. \quad (4)$$

The overall training objective of this module is :

$$\mathcal{L} = \lambda_{\text{fm}}\big(\mathcal{L}_{\text{fm}}^{E \to F} + \mathcal{L}_{\text{fm}}^{F \to E}\big) + \lambda_{\text{KL}}\mathcal{L}_{\text{KL}} + \lambda_{\text{align}}\mathcal{L}_{\text{align}} + \lambda_{\text{ROI-align}}\mathcal{L}_{\text{ROI-align}} + \lambda_{\text{recon}}\big(\mathcal{L}_{\text{recon}}^{\text{fMRI}} + \mathcal{L}_{\text{recon}}^{\text{EEG}}\big). \quad (5)$$

This formulation links EEG and fMRI latent spaces through reversible flow mappings, while reconstruction and regularization terms preserve both global and regional fidelity, thereby enhancing cross-modal consistency.

### 3.3 CROSS-MODAL ROI-WISE STRUCTURAL MODULE

To ensure that the reconstructed fMRI preserves realistic regional structural organization, we introduce an ROI-level structural constraint for fMRI data in Cross-Modal ROI-wise Structural Module,

denoted as $\mathcal{L}_{\text{ROI-struct}}$. This constraint leverages functional connectivity (FC) estimated from ground-truth EEG to regulate the pairwise relationships among fMRI ROI embeddings. Since FC reflects the strength of statistical dependencies between brain regions Rojas et al. (2018); Bijsterbosch et al. (2018), ROI embeddings with strong EEG-derived connectivity are expected to exhibit stronger associations and thus should lie closer in the fMRI embedding space, whereas those with weak connectivity should remain farther apart. In this way, the constraint enforces cross-modal consistency while grounding the embedding geometry in neurophysiological plausibility.

Formally, given the ROI-level EEG signals obtained from the electrode-to-ROI mapping described in Section 3.2, we compute the functional connectivity matrix $W \in \mathbb{R}^{N_{\text{ROI}} \times N_{\text{ROI}}}$ by first estimating the covariance across ROI time series and then normalizing each entry to obtain the Pearson correlation between ROI $r$ and ROI $r'$. The resulting normalized connectivity matrix is defined as: $\widetilde{W}_{r,r'} = \frac{W_{r,r'}}{\sum_{i<j} W_{i,j}}$, which serves as a structural prior over brain ROI interactions.

For the fMRI side, let $Z_{\text{ROI}}^{\text{fMRI}} = \{z_r^{\text{fMRI}}\}_{r=1}^{N_{\text{ROI}}}, \quad z_r^{\text{fMRI}} \in \mathbb{R}^d$ denote the ROI-level embeddings produced by the fMRI encoder. We then define the pairwise squared distance between ROI embeddings: $D_{r,r'} = \|z_r^{\text{fMRI}} - z_{r'}^{\text{fMRI}}\|_2^2$. The ROI-wise structural constraint is formulated as the weighted average of these distances:

$$\mathcal{L}_{\text{ROI-struct}} = \frac{1}{N_{\text{pairs}}} \sum_{r<r'} \widetilde{W}_{r,r'} \, D_{r,r'}, \tag{6}$$

where $N_{\text{pairs}} = N_{\text{ROI}}(N_{\text{ROI}} - 1)/2$ is the number of ROI pairs.

By constraining fMRI ROI-wise relationships based on EEG-derived connectivity, this module transfers brain structural information across modalities and encourages fMRI reconstructions that are more consistent with underlying neurophysiology.

### 3.4 R2E PHYSIOLOGICAL CONNECTIVITY GUIDANCE MODULE

To ensure that the generated fMRI retains information sufficient for reconstructing neurophysiologically plausible EEG, we constrain the *EEG covariance connectivity* of the EEG reconstructed from generated fMRI to match that of ground-truth EEG on the SPD manifold $S_d^{++}$. Let $\Sigma_{\text{GT}}^{\text{EEG}}$ and $\Sigma_{\text{Gen}}^{\text{EEG}}$ denote the ROI-level EEG *covariance* matrices computed from ground-truth EEG and from the EEG reconstructed from generated fMRI, respectively (both estimated with an OAS-type shrinkage estimator; the electrode-to-ROI operator $M$ is given in Sec. 3.3). A technical difficulty is that such covariance-based connectivity matrices live on the non-Euclidean manifold of symmetric positive-definite (SPD) matrices, where naïve Euclidean losses are geometrically inappropriate and often numerically unstable. Following the recent Riemannian-to-Euclidean (R2E) strategy Mellot et al. (2024); Collas et al. (2025), we enforce positive-definiteness and stability via symmetrization, shrinkage, and eigenvalue clamping:

$$\Sigma \leftarrow \tfrac{1}{2}(\Sigma + \Sigma^\top), \qquad \Sigma \leftarrow (1 - \lambda)\Sigma + \lambda I, \ \lambda \in (0, 1), \qquad \Lambda \leftarrow \max(\Lambda, \epsilon).$$

We then adopt the Log–Euclidean (LE) framework, which diffeomorphically maps $S_d^{++}$ to the vector space of symmetric matrices $\text{Sym}(d)$ via the matrix logarithm; using the Frobenius norm in the log-domain yields the LE distance: $d_{\text{LE}}(\Sigma_1, \Sigma_2) = \|\log(\Sigma_1) - \log(\Sigma_2)\|_F$. Our EEG covariance consistency loss is therefore:

$$\mathcal{L}_{\text{eeg-cov}} = \big\| \log(\Sigma_{\text{Gen}}^{\text{EEG}}) - \log(\Sigma_{\text{GT}}^{\text{EEG}}) \big\|_F^2. \tag{7}$$

This geometry-aware loss is applied end-to-end through the fMRI $\rightarrow$ EEG path, thereby indirectly supervising the generated fMRI to preserve essential covariance connectivity structure.

### 3.5 TRAINING STRATEGY

To achieve stable cross-modal generation, we adopt a two-stage training strategy. In the first stage, we independently train the EEG $\rightarrow$ fMRI and fMRI $\rightarrow$ EEG pathways with reconstruction and alignment losses, ensuring modality-faithful encoders/decoders ($E_{\text{EEG}}, E_{\text{fMRI}}, D_{\text{EEG}}, D_{\text{fMRI}}$) and stable bidirectional flow modules ($F_{E \rightarrow F}, F_{F \rightarrow E}$) without cycle constraints and $L_{\text{ROI-struct}}$ and $L_{\text{eeg-cov}}$.

In the second stage, we introduce cycle pathway (EEG→fMRI→EEG) to enforce reversibility of cross-modal mappings. By requiring the reconstructed EEG signals to remain consistent with their

ground truth, the cycle loss preserves shared neural information and indirectly enhances the integrity of reconstructed fMRI. The overall training objective at stage two combines all losses:

$$\mathcal{L}_{\text{total}} = \lambda_{\text{fm}}\big(\mathcal{L}_{\text{fm}}^{E \to F} + \mathcal{L}_{\text{fm}}^{F \to E}\big) + \lambda_{\text{KL}}\big(\mathcal{L}_{\text{KL}} + \mathcal{L}_{\text{ROI-KL}}\big) + \lambda_{\text{align}}\big(\mathcal{L}_{\text{align}} + \mathcal{L}_{\text{ROI-align}}\big) + \lambda_{\text{recon}}\big(\mathcal{L}_{\text{recon}}^{\text{fMRI}} + \mathcal{L}_{\text{recon}}^{\text{EEG}}\big)$$
$$+ \lambda_{\text{eeg-cov}}\mathcal{L}_{\text{eeg-cov}} + \lambda_{\text{ROI-struct}}\mathcal{L}_{\text{ROI-struct}}.$$

(8)

Here, $\mathcal{L}_{\text{fm}}$ denotes the flow matching loss, $\mathcal{L}_{\text{KL}}$ and $\mathcal{L}_{\text{ROI-KL}}$ are global and ROI-level KL loss, $\mathcal{L}_{\text{align}}$ and $\mathcal{L}_{\text{ROI-align}}$ enforce cross-modal alignment at global and ROI levels, $\mathcal{L}_{\text{recon}}$ terms regularize the fidelity of fMRI/EEG reconstructions, $\mathcal{L}_{\text{eeg-cov}}$ enforces EEG functional connectivity consistency in the Riemannian space, $\mathcal{L}_{\text{ROI-struct}}$ transfers EEG-derived structural priors to fMRI ROI embeddings. The coefficients $\lambda$ balance the contributions of different objectives.

This staged training strategy first stabilizes the cross-modal mappings by learning modality-preserving encoders, decoders, and flow modules, and then progressively incorporates structural priors and cycle-consistency constraints, thereby enabling the generation of fMRI that are not only neurobiologically faithful, but also preserve more complete neural information from EEG.

## 4 EXPERIMENT

### 4.1 SETTINGS

**Datasets**

We conduct experiments on two public EEG–fMRI datasets, NODDI Deligianni et al. (2016; 2014) and Oddball Walz et al. (2013; 2014; 2015), both providing synchronized multimodal recordings from healthy adults. The NODDI dataset contains simultaneous resting-state EEG–fMRI recordings from 15 subjects (originally 17). EEG signals were acquired using a 64-channel cap to capture millisecond-level neural dynamics, while the corresponding fMRI data were collected with a clinical 1.5T scanner. The Oddball dataset consists of recordings from 17 subjects performing auditory and visual oddball tasks. Here, EEG signals were recorded across 43 channels with high-frequency sampling to capture attentional modulations, and fMRI data were obtained using a 3T scanner to measure task-related hemodynamic responses.

**Comparison Methods**

We compare our method against six representative baselines: CNN-TC Liu & Sajda (2019), CNN-TAG Calhas & Henriques (2022), NT-ViT Lanzino et al. (2024), E2fNet Roos et al. (2025), E2fGAN Roos et al. (2025), and Spec2VolCAMU He et al. (2025), providing a diverse set of approaches for EEG-to-fMRI generation.

**Experimental Setups**

We conduct experiments on two public EEG–fMRI datasets, NODDI and Oddball. NODDI Zhang et al. (2012) comprises 15 subjects with 63-channel EEG recorded at 512 Hz over 2.16 s (1106 time points) paired with fMRI volumes of size 30×64×64, yielding about 4,500 samples. Oddball Calhas & Henriques (2022) includes 17 subjects with 34-channel EEG (512 Hz, 2.0 s; 1024 time points) paired with fMRI volumes of size 32×64×64, totaling around 17,340 samples. Raw EEG signals are transformed into Mel spectrograms using $n_{mels}$=16 and hop length of 16 samples, resulting in input sizes of [63, 16, 70] for NODDI and [34, 16, 65] for Oddball. To establish cross-modal correspondence, all fMRI volumes are parcellated using the Harvard–Oxford cortical atlas Craddock et al. (2012), and EEG electrodes are aligned to cortical regions via a fixed Biosemi64-to-ROI mapping matrix.

We report results under two validation settings: fixed split and leave-one-subject-out (LOSO) in Table 1: (1) Fixed split. Following prior work Lanzino et al. (2024), we adopt the split where a subset of subjects is used as the test set (NODDI: 2 subjects; Oddball: 4 subjects) and each experiment is repeated three times with different random seeds, and we report the mean ± standard deviation. (2) LOSO. To assess cross-subject generalization, we further conduct leave-one-subject-out (LOSO) cross-validation. In each fold, one subject is held out for testing while the remaining subjects are used for training. The final performance is obtained by averaging across all folds and reporting the corresponding standard deviations.

| Dataset | | NODDI | | | Oddball | | |
|---|---|---|---|---|---|---|---|
| Method | Validation | RMSE ↓ | SSIM ↑ | PSNR ↑ | RMSE ↓ | SSIM ↑ | PSNR ↑ |
| CNN-TC Liu & Sajda (2019) | Fixed | 0.46 ± 0.08 | 0.449 ± 0.060 | \ | 0.86 ± 0.03 | 0.189 ± 0.038 | \ |
| CNN-TAG Calhas & Henriques (2022) | Fixed | 0.40 ± 0.02 | 0.462 ± 0.020 | \ | 0.70 ± 0.09 | 0.200 ± 0.017 | \ |
| NT-ViT Lanzino et al. (2024) | Fixed | 0.07 ± 0.01 | 0.602 ± 0.005 | 23.10± 1.24 | 0.08 ± 0.012 | 0.651 ± 0.012 | 21.94±1.217 |
| E2fNet Roos et al. (2025) | Fixed | 0.17 ± 0.003 | 0.617 ± 0.003 | 15.39 ± 0.153 | 0.12 ± 0.003 | 0.64 ± 0.042 | 18.42 ± 0.220 |
| E2fGAN Roos et al. (2025) | Fixed | 0.15 ± 0.021 | 0.549 ± 0.084 | 16.48 ± 0.963 | 0.09 ± 0.01 | 0.575 ± 0.056 | 20.92 ± 0.972 |
| Spec2VolCAMU He et al. (2025) | Fixed | 0.096 ± 0.01 | 0.686 ± 0.051 | 20.35 ± 0.836 | 0.09 ± 0.03 | 0.712 ± 0.083 | 21.21 ± 3.00 |
| NeuroCycle (ours) | Fixed | **0.069±0.004** | **0.716±0.015** | **23.22±0.473** | **0.060±0.008** | **0.896±0.038** | **24.43±0.934** |
| NT-ViT Lanzino et al. (2024) | LOSO | 0.080±0.01 | 0.581±0.048 | 21.94±1.056 | 0.070±0.01 | 0.633±0.064 | 23.10±1.351 |
| E2fNet Roos et al. (2025) | LOSO | 0.102± 0.02 | 0.609± 0.062 | 19.82± 1.451 | 0.13± 0.03 | 0.626± 0.053 | 17.73± 1.825 |
| Spec2VolCAMU He et al. (2025) | LOSO | 0.088 ± 0.02 | 0.698 ± 0.068 | 21.11 ± 1.956 | 0.074 ± 0.01 | 0.742± 0.052 | 22.61± 1.373 |
| NeuroCycle (ours) | LOSO | **0.065 ± 0.009** | **0.715 ± 0.046** | **23.74 ± 1.31** | **0.061 ± 0.01** | **0.894 ± 0.053** | **24.29 ± 1.182** |

Table 1: Quantitative comparison on NODDI and Oddball datasets using RMSE, SSIM, and PSNR. Bold numbers indicate the best performance, and underlined numbers denote the second best.

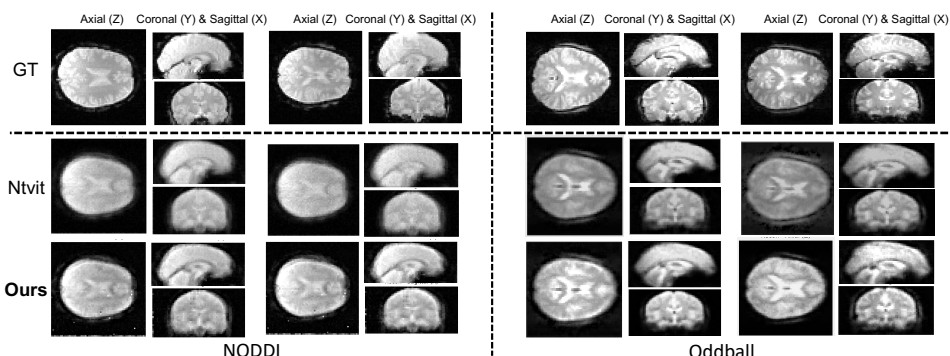

Figure 4: Three-view slices (axial, coronal, sagittal) visualization of reconstructed fMRI on NODDI and Oddball datasets.

We train the model in two stages. In the first stage, the EEG→fMRI and fMRI→EEG pathways are optimized independently for 50 epochs using Adam (lr = 0.0001, batch size = 64). In the second stage, we introduce cycle consistency and connectivity constraints, and continue training for 150 epochs with the same learning rate 0.0001. All experiments are implemented in PyTorch and run on a single NVIDIA RTX 4090 GPU.

**Training and Evaluation Metrics**

To assess the quality of the reconstructed fMRI volumes, we employ three standard image-level evaluation metrics: Root Mean Squared Error (RMSE), Structural Similarity Index (SSIM), and Peak Signal-to-Noise Ratio (PSNR). All metrics are computed in the voxel space on a per-sample basis and averaged over the test set.

## 4.2 RESULTS AND ANALYSIS

As shown in Table 1, our proposed framework (NeuroCycle) achieves the best performance on both NODDI and Oddball datasets across all metrics (RMSE, SSIM, and PSNR), consistently outperforming previous baselines. In particular, NeuroCycle reduces reconstruction error while substantially improving structural similarity and signal-to-noise ratio. Furthermore, as shown in the three-view slices (axial, coronal, sagittal) in Figure 4, our method preserves more structural and physiological details, yielding clearer and sharper reconstructions that are closer to the ground truth compared with competing approaches.

## 4.3 ABLATION STUDY

We design three ablation settings to examine the role of each component: (i) w/o Cycle, removing fMRI→EEG pathway and cycle loss, leaving only a unidirectional EEG→fMRI mapping; (ii) w/o R2E-PCG, excluding the R2E functional connectivity guidance; and (iii) w/o CM-RS, removing

| Dataset | NODDI | | | Oddball | | |
|---|---|---|---|---|---|---|
| Method \ Metric | RMSE ↓ | SSIM ↑ | PSNR ↑ | RMSE ↓ | SSIM ↑ | PSNR ↑ |
| w/o Cycle | 0.13 | 0.51 | 19.56 | 0.12 | 0.769 | 23.14 |
| w/o R2E-PCG | 0.097 | 0.64 | 20.22 | 0.074 | 0.836 | 24.28 |
| w/o CM-RS | 0.082 | 0.68 | 21.62 | 0.083 | 0.818 | 24.67 |
| NeuroCycle (ours) | **0.068** | **0.715** | **23.31** | **0.061** | **0.894** | **25.28** |

Table 2: Ablation study on NODDI and Oddball datasets. Results show that removing any component leads to performance degradation, confirming the complementary contributions of all three.

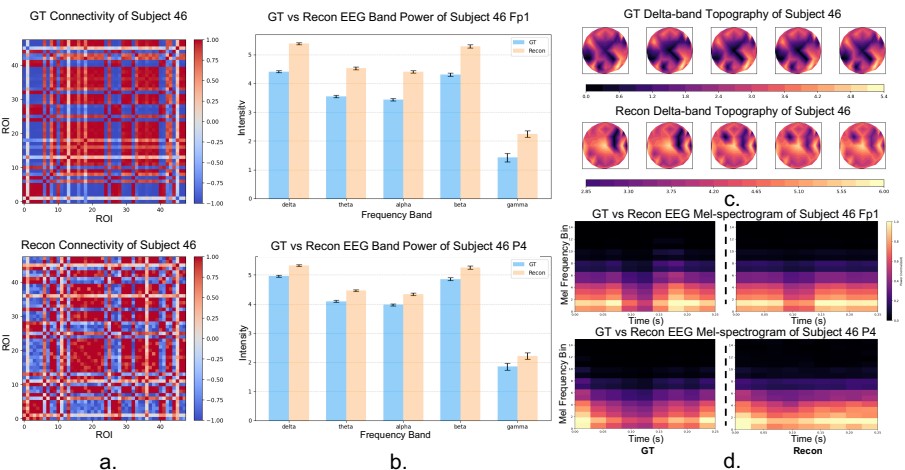

Figure 5: Visualization of reconstructed EEG versus ground truth (GT) EEG within the proposed EEG→fMRI→EEG cyclic generation framework. (a) ROI-wise functional connectivity matrices of GT and reconstructed EEG for Subject 46. (b) Band power comparison ($\delta$, $\theta$, $\alpha$, $\beta$, $\gamma$) across channels 3 and 7 for Subject 46, showing consistency between GT and reconstructed EEG. (c) Topographic maps of two major frequency bands (beta and delta) for Subject 46, where reconstructed patterns closely resemble GT spatial distributions. (d) Mel-spectrogram comparison.

the cross-modal ROI-wise structural constraint. Table 2 summarizes the overall ablation results by sequentially removing each of the three proposed components. We observe consistent performance drops across fMRI quality metrics, demonstrating that every module contributes to the overall effectiveness of our framework.

To intuitively demonstrate the effect of the cycle supervision, we examine the reconstructed EEG from four complementary perspectives—structural connectivity, spectral fidelity, spatial topography, and the Mel-spectrogram Nuwer (1988); Deligianni et al. (2014). Representative results for Subject 46 in the NODDI dataset are visualized in Fig. 5. First, we assessed structural consistency at the connectivity level. As shown in Fig. 5(a), the correlation matrix of the reconstructed EEG preserves ROI-wise connectivity patterns comparable to those of the ground truth, indicating that the cyclic pathway enforces meaningful structural organization. Second, we examined frequency fidelity. As shown in Fig. 5(b), the band power distributions on channels Fp1 and P4 demonstrate that the reconstructed EEG maintains the characteristic spectral profile across $\delta$−$\gamma$ bands, supporting the preservation of frequency-specific information. Third, we analyzed spatial distributions in terms of topographic activity. As shown in Fig. 5(c), the reconstructed signals recover coherent patterns in the major frequency band: $\beta$ and $\delta$ bands, which align with ground-truth spatial maps. And as shown in Fig. 5(d), the reconstructed EEG Mel-spectrogram preserves the dominant low-frequency components as well as the overall temporal modulation patterns.

These analyses together demonstrate that the EEG reconstructed from the generated fMRI preserves the characteristic features of the original EEG across connectivity, spectral, and spatial dimensions. This indicates that cycle supervision indirectly enforces the generated fMRI to retain complete neu-

ral information, thereby providing strong evidence for the effectiveness and rationality of the proposed framework in improving fMRI generation quality.

## 5 CONCLUSION

In this work, we proposed a cyclic EEG–fMRI generation framework that integrates neurophysiological priors to ensure information completeness and neuroscientific plausibility. The Cross-Modal ROI-wise Structural Module preserves ROI-level organization, and the R2E Physiological Connectivity Guidance Module enforces functional connectivity consistency. The cyclic bidirectional design further guarantees preservation of essential neural information. Experiments on two public datasets demonstrate superior fMRI reconstruction with richer neural information and improved cross-modal consistency, providing a principled approach for interpretable and clinically relevant brain signal generation.

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

## A  Appendix

### A.1  LLM Usage

Large Language Models (LLMs) were used to support the preparation of this manuscript. In particular, an LLM was employed to assist with refining the paper, focusing on improving readability, polishing grammar, and ensuring clarity and conciseness. The model also provided alternative phrasings and stylistic adjustments to better align with academic writing conventions.

The LLM was not involved in research ideation, methodological design, experimental implementation, or result analysis. All scientific contributions including conceptual development, data processing, model design, and evaluation were carried out exclusively by the authors. The role of the LLM was limited to linguistic refinement and presentation quality.

The authors take full responsibility for the manuscript's scientific content, including any portions of text that were edited or polished using the LLM. Care has been taken to ensure that the use of the LLM adheres to ethical standards and does not introduce plagiarism or scientific misconduct.

