# OpenReview forum: "NeuroCycle: Physiologically Constrained Cycling for Generating Neural Information-Rich fMRI from EEG"
_ICLR.cc/2026/Conference — Submitted to ICLR 2026_

### Official Review · Reviewer_obju · 2025-10-27

**Soundness:** 2
**Presentation:** 2
**Contribution:** 1
**Rating:** 2
**Confidence:** 4

**Summary:**

The paper proposes an approach to neural transcoding from the EEG to the fMRI modality. Among all, the authors propose a ROI approach which is unique to this particular task, as well as the clever idea of implementing an EEG to fMRI to again EEG cycle to improve the stability of the generated data.

**Strengths:**

- the paper presents a different take on this task wrt the current SotA, by proposing to align both EEG and fMRI features to physical positions in the head.

- performances are SotA, as per Tab. 1.

- qualitative results are also provided in Fig. 4, directly compared with another method in literature.

**Weaknesses:**

- the flow of the approach is not entirely clear. Sec. 3.2, particularly, seems unnecessarily complicated with inline formulas. It distracts the reader from following the flow of the approach.

- generated images, as per Fig. 4, looks very similar to NT-ViT for me. Just like NT-ViT, they fail to capture the correct shape of the brain and more complex inner structures, especially visible to me on the Sagittal view of the first example of the NODDI dataset.

- the cross-validation scheme used for the experiments is not specified. For instance, NT-ViT oes both kfold and LOSO cross-validations. to compare apple to apples, it is necessary to use the same protocol, and this is a very major issue (!!!).

- ablation study in Tab. 2 is partial. First, it is not compared to a baseline (no cycle, no R2E-PCG, no CM-RS), making it impossible to assess the relevance of each module. Second, combinations are not tried (like no cycle but both modules, or cycle and just CM-RS).

- in Fig. 5, the reconstructed EEG Mel-spectrogram, which would have been very informative, is not shown.

- in Tab. 1, the second-best results are not highlighted (like for example underlining it).

- there are no references or links to anonymized code in the main manuscript (!!!).

Now, as sidenotes:

- citation at L052 might be wrong as E2fGAN has not been proposed in Calhas' paper.

- on L165, the reviewer believes that "incorporate" is not the right word, because "spectral information" are already present in the signals. Maybe the authors meant something like "extract"?

- typo on L199: "...regions, We apply a..."

**Questions:**

- What naturally comes after the weaknesses discussion is that it would be good to know the validation scheme used during the experiments.

But, anyway, the reviewer believes both k-fold and LOSO results should have been presented by the authors.

- What are the practical differences, minutes and details produced by data generated with NeuroCycle wrt NT-ViT? It is hard to tell from Fig. 4 alone, and its caption is not informative enough for the reviewer to guess.

---

> ### Author Response · Authors · 2025-11-21
> **Response to Reviewer obju (Part 1)**
>
> We sincerely appreciate the reviewers’ constructive and thoughtful feedback. Your comments have greatly helped us refine the paper. We have performed the required additional experiments, addressed each point in detail below.
>
> > Weakness 1: Complicated for Sec.3.2
>
> We sincerely thank the reviewer for pointing out this issue.
> We agree that the inline formulas in Sec. 3.2 may interrupt the narrative flow and make the approach harder to follow. In the revised manuscript, we will significantly simplify this section by presenting a clearer high-level description of the pipeline and moving the detailed mathematical parts to the Appendix.
> We apologize for the inconvenience caused and appreciate the reviewer’s helpful feedback on improving the clarity of our presentation.
>
> > Weakness 2: The results look similar compared to ntvit.
>
> We thank the reviewer for the careful inspection of Fig. 4.
> We agree that, at this resolution, the overall brain silhouette may appear similar between NT-ViT and our method. This is maybe because our model is not designed to explicitly optimize global anatomical contours. Instead, our objective focuses on reconstructing physiologically meaningful fMRI patterns, such as region-level activations captured by our Cross-Modal ROI-wise Structural Module. These differences, while functionally important, are often not visually salient in slice-based views, even though they lead to clear improvements in quantitative and neuroscientifically relevant metrics.
>
> To address this concern and to more comprehensively demonstrate that the fMRI generated by NeuroCycle differs substantially from NT-ViT in terms of physiological fidelity, we evaluate the reconstructed volumes at three complementary levels, ranging from fine-grained regional patterns to large-scale functional organization and downstream task utility:
>
> (1) **ROI-level activation consistency**
>
> At the regional level, we evaluate whether the synthesized fMRI preserves the voxel-averaged activation patterns defined by anatomical ROIs. For each sample, both the predicted and ground-truth fMRI volumes are z-scored within the brain mask. We then project each 3D volume onto 71 cortical ROIs using the Harvard–Oxford atlas, resulting in one ROI activation vector per volume.
>
> Across all datasets and evaluation protocols including both the fixed split and the more stringent LOSO setting, NeuroCycle consistently achieves markedly higher ROI-wise Pearson correlation with ground truth than NT-ViT. This indicates that our method more faithfully reconstructs fine-grained regional activation patterns and preserves local functional organization that is essential for meaningful downstream analysis.
>
> Table 1. ROI-wise functional pattern consistency (mean ± std).
> Higher correlation indicates better preservation of region-level functional organization.
>
> | Dataset | Method     | ROI Corr $\rho_{\text{ROI}}$ ↑         |
> |---------|------------|-----------------------|
> | NODDI (fixed split)   | NT-ViT     | 0.9104 ± 0.1326          |
> | NODDI  (fixed split) | NeuroCycle | **0.9622 ± 0.0017**      |
> | NODDI (LOSO)   | NT-ViT     | 0.9275 ± 0.0928          |
> | NODDI  (LOSO) | NeuroCycle | **0.9657 ± 0.0008**      |
> | Oddball  (fixed split)  | NT-ViT   | 0.8967 ± 0.0136    |
> | Oddball  (fixed split) | NeuroCycle | **0.9173 ± 0.0055**      |
> | Oddball   (LOSO) | NT-ViT  | 0.9127 ± 0.0186          |
> | Oddball   (LOSO) | NeuroCycle | **0.9563 ± 0.0092**    |

---

> ### Author Response · Authors · 2025-11-21
> **Response to Reviewer obju (Part 2)**
>
> (2) **Network-level functional organization**
>
> At the network level, we project each generated fMRI volume onto the Yeo-7 atlas to assess whether large-scale functional organization is preserved. For each subject and each sample, we compute network-level activation profiles by averaging voxel responses within each Yeo network, and then measure the correlation between the generated and ground-truth profiles. Across both datasets under LOSO evaluation, NeuroCycle achieves consistently higher correlations with ground-truth network activations than NT-ViT, both overall and within individual networks. The improvements are especially clear in higher-order systems such as the dorsal attention, frontoparietal, and default-mode networks. These results show that NeuroCycle better recovers global functional structure and inter-regional dependencies that are not apparent from slice-based visual observation alone.
>
> **Table 2. Network-level spatial organization consistency using Yeo functional networks. **
>
> | Dataset | Method | All Networks $\rho$ ↑ | Visual $\rho$ ↑ | Somatomotor $\rho$ ↑ | Dorsal Attention $\rho$ ↑ | Salience $\rho$ ↑ | Limbic $\rho$ ↑ | Frontoparietal $\rho$ ↑ | DMN $\rho$ ↑ |
> |------|---|----|-------|-----|------|--------|---------|-------|------|
> | NODDI (LOSO)  | NT-ViT    | 0.928 ± 0.041  | 0.827 ± 0.020  | 0.776 ± 0.053 | 0.725 ± 0.025    | 0.820 ± 0.027         | 0.806 ± 0.037         | 0.822 ± 0.033          | 0.854 ± 0.030         |
> | NODDI (LOSO) | NeuroCycle | 0.955 ± 0.030  | 0.953 ± 0.022   | 0.929 ± 0.041         | 0.929 ± 0.042          | 0.944 ± 0.021         | 0.900 ± 0.043         | 0.931 ± 0.030 | 0.949 ± 0.018    |
> | Oddball  (LOSO) | NT-ViT   | 0.929 ± 0.058   | 0.856 ± 0.063       | 0.814 ± 0.046  | 0.829 ± 0.053       | 0.861 ± 0.025       | 0.879 ± 0.032       | 0.878 ± 0.068       | 0.906 ± 0.041   |
> | Oddball  (LOSO) | NeuroCycle | 0.972 ± 0.023  | 0.923 ± 0.030   | 0.879 ± 0.032         | 0.854 ± 0.072          | 0.928 ± 0.023         | 0.917 ± 0.032         | 0.931 ± 0.021  | 0.942 ± 0.018         |
>
> (3) **Downstream cognitive-task performance**
>
> Also, we verify whether the reconstructed fMRI carries functionally meaningful information that supports real analysis tasks. We trained a lightweight 3D CNN classifier (~2.9×10⁵ parameters) to discriminate between the two Oddball conditions (task001 vs. task002) using either ground-truth or synthesized fMRI volumes. The classifier was trained under identical settings for both inputs. Results show that the model achieves comparable accuracy on generated and real fMRI, demonstrating that NeuroCycle preserves the functional patterns necessary for downstream cognitive-state decoding.
>
> | Input Type       | Test Accuracy |
> |-------|---------|
> | Generated fMRI of NT-ViT  | 91.73% |
> | Generated fMRI of Ours (NeuroCycle)  | **98.42%** |
> | Real fMRI       | 99.92% |
>
> Together, these three levels of analysis ranging from local ROI activations to global network organization and downstream task utility, provide evidence that NeuroCycle generates physiologically meaningful and functionally interpretable fMRI, even when such differences are not easily visible from slice-based inspection alone.
>
> We will incorporate these results into the revised manuscript, and we hope that they address the reviewer’s concerns. If there are any additional questions or points that require further clarification, we would be happy to provide more details.
>
> > Weakness 3: Cross-validation scheme.
>
> We thank the reviewer for highlighting this important issue. We apologize for not clearly specifying the cross-validation protocol in the original submission. Our main experiments were conducted using the fixed split setting, following the configuration used in NT-ViT’s primary experiments.
>
> To ensure a fair, apples-to-apples comparison, we have now additionally performed LOSO cross-validation, consistent with the protocol adopted in NT-ViT. Specifically, for the i-th fold, the i-th subject is held out as the test set, and the remaining subjects are used for training. The new LOSO results are reported below and will be incorporated into the revised manuscript.
>
> **NODDI dataset (15 subjects) LOSO evaluation:** We conducted LOSO across all 15 subjects; the results is provided below:
>
> | Method |  RMSE |SSIM | PSNR |
> |-----|------|------|------|
> | NTVIT(LOSO)  | 0.080 $\pm$ 0.01     | 0.581$\pm$0.048    | 21.94$\pm$1.056
> | NeuroCycle(LOSO)  |  **0.065$\pm$0.009**| **0.715$\pm$0.046**| **23.74$\pm$1.31**  |
>
> **Oddball dataset (17 subjects) LOSO evaluation:** Similarly, LOSO was performed across all 17 subjects; the results are shown below:
> | Method |  RMSE |SSIM | PSNR |
> |-----------|------|------|------|
> | NTVIT(LOSO)  |  0.07$\pm$0.01      | 0.633$\pm$0.064    | 23.10$\pm$1.351    |
> | NeuroCycle(LOSO)  |  **0.061$\pm$0.01** | **0.894$\pm$0.053**| **24.29$\pm$1.182**|
>
> Across both datasets, NeuroCycle consistently outperforms NT-ViT under LOSO scheme.

---

> > ### Author Response · Authors · 2025-11-21
> > **Response to Reviewer obju (Part 3)**
> >
> > > Weakness 4: More ablation studies
> >
> > We thank the reviewer for this helpful comment.
> > Our original ablation focused on leave-one-out settings (w/o Cycle, w/o R2E-PCG, w/o CM-RS). For clarity, we restate the relationship among the components. In our design, the Cycle branch is required for enabling the R2E-PCG module, because the physiological connectivity supervision in R2E-PCG relies on the fMRI→EEG pathway introduced by the cycle. Therefore, the ablation setting w/o Cycle is functionally equivalent to the configuration where only CM-RS is used.
> >
> > To more clearly show the contribution of our method, we additionally include a full baseline without any of the three modules (no Cycle, no R2E-PCG, no CM-RS). The results of this baseline have now been added and are shown below, and will be incorporated into the revised manuscript.
> >
> > | Method / Metric | **NODDI RMSE ↓** | **NODDI SSIM ↑** | **NODDI PSNR ↑** | **Oddball RMSE ↓** | **Oddball SSIM ↑** | **Oddball PSNR ↑** |
> > |-----------------|------------------|-------------------|-------------------|---------------------|----------------------|----------------------|
> > | w/o All  | 0.156  | 0.474    | 19.15  | 0.137 | 0.753    | 23.11   |
> > | w/o Cycle     | 0.13  | 0.51  | 19.56 | 0.12  | 0.769 | 23.14 |
> > | w/o R2E-PCG   | 0.097 | 0.64  | 20.22 | 0.074 | 0.836 | 24.28 |
> > | w/o CM-RS     | 0.082 | 0.68  | 21.62 | 0.083 | 0.818 | 24.67 |
> > | NeuroCycle (ours) | 0.068  | 0.715 | 23.31| 0.061 | 0.894| 25.28     |
> >
> > > Weakness 5: the reconstructed EEG Mel-spectrogram
> >
> > Thank you for pointing this out. We have now added the visualization of the reconstructed EEG Mel-spectrogram in Fig. 5.
> >
> > > Weakness 6: Highlighting the second-best results
> >
> > We have corrected this in the main manuscript.
> >
> > > Weakness 7: The release of codes
> >
> > We fully agree that code availability is important. Upon acceptance, we will release the complete codebase.
> >
> > > sidenotes：
> >
> > The citation, spelling, and wording issues have now been corrected in the main manuscript.
> >
> > > Questions
> >
> > In the revised manuscript, we have added a clear description of the validation scheme and now include both fixed-split and LOSO results as suggested.
> >
> > Regarding the practical differences between NeuroCycle and NT-ViT, we agree that Fig. 4 alone may not fully reflect them. As shown in the response for Weakness 2, NeuroCycle consistently produces fMRI volumes with higher physiological fidelity, outperforming NT-ViT at three levels:
> > (1) more accurate ROI-level activation patterns,
> > (2) better preservation of large-scale functional networks, and
> > (3) stronger performance on downstream cognitive-state decoding.
> >
> > These results show that NeuroCycle generates fMRI that is not only visually plausible but also functionally meaningful.

---

### Official Review · Reviewer_4Cgs · 2025-10-28

**Soundness:** 3
**Presentation:** 3
**Contribution:** 3
**Rating:** 6
**Confidence:** 3

**Summary:**

The paper proposes a method (NeuroCycle) to generate fMRI data from EEG recordings. The main contribution of the paper over the previous methods is to try and enforce neurophysiological constraints on the generated fMRI data. The authors claim to achieve this by adding two modules on top of VAE-style EEG/fMRI encoder/decoder modules (the two resulting representations are themselves also connected through a bidirectional cross-"flow" matching module). The first one (named cross-modal ROI-wise structural module) is meant to preserve the functional connectivity observed in the GT-EEG data in the learned fMRI embeddings (which are used to reconstruct EEG data in the next step). The idea being that ROIs that have stronger functional connections should have embeddings that map closer to each other in the embedding space. The second module (named R2E Physiological connectivity guidance module) uses the GT-EEG covariance connectivity and attempts to make sure that it is similar to the reconstructed EEG covariance connectivity with distance calculated using log-euclidean distance. The method is "cyclical" in the sense that the GT-EEG data is used to generate fMRI data, the encoded embeddings from this fMRI data in turn reconstruct EEG data with neurophysiological constraints enforced through the above mentioned modules.

**Strengths:**

The paper tackles an important problem in neuroscience i.e. the feasibility of obtaining fMRI data vs the relative ease with which EEG data can be obtained. It is a significant problem as it's nearly impossible to obtain fMRI data for research purposes in developing countries (in some countries there is a single site available for the whole country). If high-quality neurophysiologically relevant fMRI data can be generated from EEG, that can be impactful.

The overall idea is strong. Without constraints it is indeed likely that fMRI data generated from EEG would look visually plausible but may not make neuroscientific sense.

The paper is also generally well-written and laid out and is easy to follow.

**Weaknesses:**

- My first concern about the method is the number of hyperparameters involved. There are 6 hyperparameters that would need to be tuned just in the loss function. Not to mention the architecture and training parameters involved in the training and the temperature parameter.
- The experiments section is light on details - it's not immediately clear how the data was divided into train/val/test (if at all). There's also no mention of how and what values of hyperparameters were chosen. Also, what does generalization look like? Does the model generalize only within datasets? Does it generalize outside datasets? Can I train on oddball and test on NODDI? Other way round? Or do I have to retrain on a portion of my EEG dataset everytime? which would reduce the applicability of the method.
- It's also unclear why the cross-modal "flow matching" module is called as such. As far as I can tell there's no flow-matching happening. The only thing that is happening is that the MSE between the EEG and fMRI embeddings is being minimized in cross-spaces. I don't think that constitutes flow matching, you'd need some notion of velocity of these embeddings for that to be happening here, which could be there if there was a notion of time in these embeddings. Which, as far as I can tell, isn't there.
- Also, please put citations in parenthesis at end of sentences, will improve readability a lot.

**Questions:**

- I'm curious to know what happens to time? is it just averaged out at the very beginning (line 208)? so the generated fMRI images are just mean images?
- Figure 4, while yes, the proposed method does generate sharper images, I'm not convinced by those particular images that the generated fMRI data is similar to the GT. Especially so in the case of Oddball. It might be worth including fMRI expert evaluations on random subsets to ensure the useful aspects of the data are actually being generated.
- Figure 5, I wonder what these would look like if you take the generated fMRI data from competitor methods and generate EEG data from those on a trained model from your method.

---

> ### Author Response · Authors · 2025-11-24
> **Response to Reviewer 4Cgs (Part 1)**
>
> Thank you for your positive feedback and constructive suggestions. They are very encouraging to us. Below, we provide our responses and hope to address your questions.
>
> > Weakness 1: Details of hyperparameters
>
> Although the loss contains six weighting coefficients, we found that the training is robust to a wide range of choices. We followed a two-stage strategy: (i) a coarse grid search on the NODDI training split to determine stable ranges, and (ii) fixing these values for all experiments.
>
> Importantly, the exact same hyperparameters were used for both datasets and for all LOSO folds, without any re-tuning. In the LOSO setting, the i-th subject is held out as the test set while all remaining subjects are used for training, and we intentionally did not adjust any hyperparameters for each fold. The consistent performance across all LOSO evaluations further demonstrates that the method is highly insensitive to hyperparameter choices.
>
> > Weakness 2: Dataset split
>
> We apologize for the missing details. For the main experiments, we adopted a **fixed split** in which two subjects were randomly selected as the test set, and all remaining subjects were used for training. To further ensure evaluation fairness and generalization, we additionally conducted a full **leave-one-subject-out (LOSO)** cross-validation.
> In LOSO, for each fold i, the i-th subject is held out entirely as the test set, while all other subjects are used for training, with no change of hyperparameters. We performed this procedure for all subjects in the dataset.
>
> We have add the experiment results and dataset descriptions in the revised manuscript.
>
> > Weakness 2: The generalization
>
> Our experiments focus on within-dataset generalization, and we evaluate this thoroughly using a full leave-one-subject-out (LOSO) cross-validation.
> In LOSO, each subject is held out as the test set once, and the model is trained on all remaining subjects without re-tuning hyperparameters across folds.
> The stable performance across all LOSO folds demonstrates that NeuroCycle generalizes well to unseen subjects within the same dataset.
>
> Regarding cross-dataset generalization (e.g., training on Oddball and testing on NODDI), direct training on Oddball and testing on NODDI is not feasible because the datasets differ substantially in: EEG channel montage、TR and temporal sampling、fMRI preprocessing pipeline, e.t.c.
>
> These differences result in incompatible input–output spaces, making zero-shot cross-dataset application much harder. This is a limitation shared by current EEG→fMRI or fMRI→EEG generation frameworks. Nevertheless, we view this as an important and promising direction for future research.
>
> > Weakness 2: Concern About the Need for Retraining
>
> Yes, NeuroCycle must be trained on part of the target EEG–fMRI dataset before use.
> This is a common paradigm in multimodal neuroimaging, as neural signals exhibit substantial domain shifts across datasets due to differences in acquisition protocols, electrode configurations, and task paradigms, making it inherently difficult to define a unified input space across studies.
>
> However, our LOSO evaluation demonstrates that once trained under a given experimental setting, the model generalizes reliably to entirely unseen individuals, without any hyperparameter adjustment.
> This suggests that NeuroCycle remains practically usable, as it generalizes well to new subjects under the same experimental setting without the need for additional tuning.
>
> > Weakness 3: The name of flow matching
>
> We thank the reviewer for raising this point. We acknowledge that this differs from the continuous-time, ODE-based flow matching approaches that define explicit vector fields over trajectories. Instead, our method follows the discrete flow-matching formulation proposed by Lipman[1], which defines flow matching as learning a transport map between two distributions by minimizing an optimal matching loss and without requiring a continuous-time parameterization or explicit velocity fields. This formulation has recently been widely adopted in generative modeling tasks, including FlowTok[2].
>
> In our setting, flow matching reduces to minimizing an alignment loss between sample embeddings drawn from EEG and fMRI modalities. This aligns precisely with the discrete flow-matching framework, where the objective is to match cross-modal distributions via latent representation alignment.
>
> [1] Lipman Y, Chen R T Q, Ben-Hamu H, et al. Flow matching for generative modeling[J]. arXiv preprint arXiv:2210.02747, 2022.
>
> [2] He J, Yu Q, Liu Q, et al. Flowtok: Flowing seamlessly across text and image tokens[J]. arXiv preprint arXiv:2503.10772, 2025.
>
> > Weakness 4: Citation
>
> We appreciate the suggestion. We will revise the manuscript to place all citations in parentheses at the end of sentences to improve clarity and readability.

---

> ### Author Response · Authors · 2025-11-24
> **Response to Reviewer 4Cgs (Part 2)**
>
> > Question 1: Temporal Information Clarification
>
> We clarify that we do not average the EEG signals over time.
> For each fMRI volume, we extract a short EEG segment centered on the corresponding acquisition window (typically 2–4 seconds). This preserves the temporal dynamics and spectral structure of the EEG signal.
> The EEG encoder processes this segment to obtain a latent representation, which is then used to generate the fMRI volume at that specific TR.
>
> Therefore, the generated fMRI is not a “mean image”, but rather the spatial activation pattern corresponding to the specific fMRI acquisition time point.
>
> > Question 2: More convincing evaluations for fMRI reconstruction
>
> Although Figure 4 provides an intuitive comparison, the visual differences between fMRI volumes are often subtle and may not be easily distinguishable from a small number of 2D anatomical slices, particularly in datasets like Oddball where activation contrasts are inherently weak. The reason is our method focuses on preserving physiological and functional patterns, which are not always visually prominent in raw slice views.
>
> To address this concern and to more comprehensively demonstrate that the fMRI generated by NeuroCycle preserves meaningful brain patterns beyond visual appearance, we performed three complementary evaluations, covering fine-grained regional structure, large-scale functional organization, and downstream cognitive relevance:
>
> (1) **ROI-level activation consistency**
>
> At the regional level, we evaluate whether the synthesized fMRI preserves the voxel-averaged activation patterns defined by anatomical ROIs. For each sample, both the predicted and ground-truth fMRI volumes are z-scored within the brain mask. We then project each 3D volume onto 71 cortical ROIs using the Harvard–Oxford atlas, resulting in one ROI activation vector per volume.
>
> Across all datasets and evaluation protocols including both the fixed split and the more stringent LOSO setting, NeuroCycle consistently achieves markedly higher ROI-wise Pearson correlation with ground truth than NT-ViT. This indicates that our method more faithfully reconstructs fine-grained regional activation patterns and preserves local functional organization that is essential for meaningful downstream analysis.
>
> Table. ROI-wise functional pattern consistency (mean ± std).
> Higher correlation indicates better preservation of region-level functional organization.
>
> | Dataset | Method | ROI Corr $\rho_{\text{ROI}}$ ↑   |
> |--|---|---|
> | NODDI (fixed split)   | NT-ViT | 0.9104 ± 0.1326  |
> | NODDI  (fixed split) | NeuroCycle | **0.9622 ± 0.0017** |
> | NODDI (LOSO)   | NT-ViT     | 0.9275 ± 0.0928|
> | NODDI  (LOSO) | NeuroCycle | **0.9657 ± 0.0008** |
> | Oddball  (fixed split)  | NT-ViT   | 0.8967 ± 0.0136 |
> | Oddball  (fixed split) | NeuroCycle | **0.9173 ± 0.0055**  |
> | Oddball   (LOSO) | NT-ViT  | 0.9127 ± 0.0186 |
> | Oddball   (LOSO) | NeuroCycle | **0.9563 ± 0.0092** |
>
> (2) **Network-level functional organization**
>
> At the network level, we project each generated fMRI volume onto the Yeo-7 atlas to assess whether large-scale functional organization is preserved. For each subject and each sample, we compute network-level activation profiles by averaging voxel responses within each Yeo network, and then measure the correlation between the generated and ground-truth profiles. Across both datasets under LOSO evaluation, NeuroCycle achieves consistently higher correlations with ground-truth network activations than NT-ViT, both overall and within individual networks. The improvements are especially clear in higher-order systems such as the dorsal attention, frontoparietal, and default-mode networks. These results show that NeuroCycle better recovers global functional structure and inter-regional dependencies that are not apparent from slice-based visual observation alone.
>
> Table. Network-level spatial organization consistency using Yeo functional networks.
> Correlation (mean ± std) is computed between ground-truth and generated fMRI network activations.
> Higher values indicate better preservation of large-scale functional organization.
>
> | Dataset | Method | All Networks $\rho$ ↑ | Visual $\rho$ ↑ | Somatomotor $\rho$ ↑ | Dorsal Attention $\rho$ ↑ | Salience $\rho$ ↑ | Limbic $\rho$ ↑ | Frontoparietal $\rho$ ↑ | DMN $\rho$ ↑ |
> |---|---|---|---|--|---|----|----|--|----|
> | NODDI (LOSO)  | NT-ViT    | 0.928 ± 0.041| 0.827 ± 0.020  | 0.776 ± 0.053| 0.725 ± 0.025| 0.820 ± 0.027| 0.806 ± 0.037 | 0.822 ± 0.033 | 0.854 ± 0.030  |
> | NODDI (LOSO) | NeuroCycle | 0.955 ± 0.030| 0.953 ± 0.022| 0.929 ± 0.041| 0.929 ± 0.042 | 0.944 ± 0.021| 0.900 ± 0.043| 0.931 ± 0.030  | 0.949 ± 0.018 |
> | Oddball  (LOSO) | NT-ViT   | 0.929 ± 0.058  | 0.856 ± 0.063| 0.814 ± 0.046| 0.829 ± 0.053| 0.861 ± 0.025 | 0.879 ± 0.032  | 0.878 ± 0.068 | 0.906 ± 0.041 |
> | Oddball  (LOSO) | NeuroCycle | 0.972 ± 0.023| 0.923 ± 0.030 | 0.879 ± 0.032| 0.854 ± 0.072| 0.928 ± 0.023 | 0.917 ± 0.032| 0.931 ± 0.021 | 0.942 ± 0.018 |

---

> > ### Author Response · Authors · 2025-11-24
> > **Response to Reviewer 4Cgs (Part 3)**
> >
> > (3) **Downstream cognitive-task performance**
> >
> > Also, we verify whether the reconstructed fMRI carries functionally meaningful information that supports real analysis tasks. We trained a lightweight 3D CNN classifier (~2.9×10⁵ parameters) to discriminate between the two Oddball conditions (task001 vs. task002) using either ground-truth or synthesized fMRI volumes. The classifier was trained under identical settings for both inputs. Results show that the model achieves comparable accuracy on generated and real fMRI, demonstrating that NeuroCycle preserves the functional patterns necessary for downstream cognitive-state decoding.
> >
> > | Input Type       | Test Accuracy |
> > |------------------|-------------------|
> > | Generated fMRI of NT-ViT  | 91.73% |
> > | Generated fMRI of Ours (NeuroCycle)  | **98.42%** |
> > | Real fMRI       | 99.92% |
> >
> > Together, these three levels of analysis from local ROI activations to global network organization and downstream task utility demonstrate that NeuroCycle preserves physiologically meaningful and functionally interpretable fMRI patterns, even when such differences are not visually prominent in slice-based inspections.
> >
> > > Question 3. Using Competitor fMRI as Input to evaluate
> >
> > We fed fMRI generated by NT-ViT into our frozen fMRI→EEG branch and compared the reconstructed EEG with that obtained from ground-truth and NeuroCycle-generated fMRI.
> >
> > Since EEG characteristics are best evaluated quantitatively rather than visually, we report the band-power correlation, which is also the primary metric used in Fig. 5. Band-power provides a direct measure of whether the neural frequency structure is preserved when different fMRI sources are fed into our fMRI to EEG branch, making it the most relevant indicator for this specific experiment. To quantify this, we computed band-power metrics across all 63 channels and all five canonical frequency bands (δ–γ) in the NODDI dataset and reported below:
> >
> >
> > | Dataset | Method | Band-Power RMSE ↓ | Band-Power Pearson r ↑ |
> > |---------|--------------|---------------------|--------------------------|
> > | NODDI   | NT-ViT       |     2.349       |       0.381         |
> > | NODDI   | NeuroCycle(ours)       | **1.561**           | **0.585**     |
> >
> > We observed that EEG reconstructed from NT-ViT–generated fMRI shows clearly reduced spectral fidelity, reflected by higher band-power RMSE and lower correlation. In contrast, NeuroCycle-generated fMRI yields much lower spectral error (1.561 vs. 2.349) and higher cross-band correlation (0.585 vs. 0.381), approaching the quality obtained using real fMRI. This confirms that NeuroCycle preserves neural frequency structure more effectively than baseline methods.

---

### Official Review · Reviewer_DdHL · 2025-10-30

**Soundness:** 4
**Presentation:** 4
**Contribution:** 3
**Rating:** 8
**Confidence:** 2

**Summary:**

This paper proposes NeuroCycle, a cyclic EEG–fMRI generation framework that integrates neurophysiological priors to ensure information completeness and biological plausibility. Unlike previous unidirectional EEG to fMRI models, NeuroCycle employs a bidirectional cycle (EEG to fMRI to EEG) enforced through flow matching. Two key modules, i.e., (1) the Cross-Modal ROI-wise Structural Module and (2) the R2E Physiological Connectivity Guidance Module, inject physiological constraints into the model, preserving functional connectivity and ROI-level structure.

**Strengths:**

I find this paper technically novel and conceptually interesting. The bidirectional generation between EEG and fMRI addresses an important and challenging problem in multimodal neuroimaging. Unlike traditional unidirectional approaches, this work introduces a cyclic generation framework and effectively incorporates spatial priors to guide the learning of fMRI representations. The experimental validation is thorough, and the reported improvements over existing baselines are consistent and significant.

**Weaknesses:**

Currently, the paper uses regional information of EEG to guide the fMRI learning. May be one can also consider using BOLD delay characteristics to guide the learning of fMRI. Besides, since the training involves reconstructing EEG from fMRI, what is the computational complexity compared to unidirectional generation?

**Questions:**

1. Could the framework be extended to MEG or intracranial EEG, which have higher spatial precision?

2. Is the learned fMRI representation temporally aligned with the BOLD delay characteristics?

3. What is the difference between the proposed method and cycleGAN, and what is the advantage of the proposed approach in the considered scenario?

---

> ### Author Response · Authors · 2025-11-25
> **Response to Reviewer DdHL**
>
> We sincerely appreciate your positive assessment of our work and it's truly encouraging for us. Below we provide detailed responses to your questions, and we hope these clarifications could address your concerns.
>
> > Weakness: Temporal fMRI modeling
>
> We thank the reviewer for this insightful comment. The suggestion of incorporating BOLD-delay characteristics is highly relevant in the context of temporal fMRI modeling; however, our work focuses on voxel-wise 3D fMRI volume generation. BOLD delay reflects the temporal convolution of the HRF and becomes meaningful only when synthesizing 4D time-series, rather than static 3D volumes. Therefore, HRF-aligned constraints are not directly applicable to our task formulation. Instead, we adopt EEG regional representations as guidance because they provide stable, cross-modal cues that are directly aligned with the spatial nature of our task.
>
> We think your suggestion points to an interesting future direction. If the framework is extended to 4D fMRI synthesis, HRF-informed priors or voxel-wise BOLD-lag maps could be incorporated to better model temporal hemodynamic dynamics.
>
> > Weakness: Computational complexity of the bidirectional framework
>
> Compared to the unidirectional EEG→fMRI baseline (EEG encoder + fMRI decoder), the full NeuroCycle adds an fMRI encoder and an EEG decoder to realize the fMRI→EEG path. This increases the training-time complexity by roughly a factor of ≈1.6, but remains in the same order of magnitude, and the inference cost for EEG→fMRI generation remains unchanged since the backward path is only used during training.
>
> Importantly, this additional computation enables learning cycle-consistent latent spaces, which substantially improves the stability and identifiability of the cross-modal mapping which cannot be achieved with a unidirectional design.
>
> > Question 1: Extensibility to MEG or intracranial EEG
>
> We thank the reviewer for this thoughtful question. In principle, the framework can be extended to MEG or intracranial EEG, as the model operates on latent neural representations. The main adaptation for MEG or intracranial EEG would lie in the preprocessing pipeline: MEG requires transforming sensor-level signals into cortical source estimates, while intracranial EEG involves aligning electrode implantation sites to atlas-based ROIs, resulting in a sparse but spatially precise representation. The core architecture would remain largely unchanged once modality-specific representations are obtained. We view this as a promising direction for future work.
>
> > Question 2: BOLD-delay characteristics in fMRI
>
> We appreciate the reviewer’s insightful question. Our framework focuses on generating single-frame 3D fMRI volumes, and therefore it does not model temporal dynamics or the hemodynamic delay associated with the BOLD response. The learned fMRI representations are aligned with the spatial activation patterns present in each volume, rather than the temporal evolution of the BOLD signal.
>
> Since BOLD-delay characteristics become relevant only in 4D fMRI time-series reconstruction, they are not applicable to the static voxel-wise generation task considered in this work. Nevertheless, incorporating HRF-informed temporal constraints represents an interesting direction for extending the framework to time-resolved fMRI synthesis in future work.
>
> > Question 3: Difference from CycleGAN and advantages of our framework
>
> Our method is fundamentally different from CycleGAN. CycleGAN performs adversarial image-to-image translation between two visually similar domains, relying on GAN losses and pixel-level cycle consistency. In contrast, EEG and fMRI are heterogeneous neurophysiological modalities with different dimensionalities, noise characteristics, and biophysical meanings, making adversarial domain translation inappropriate.
>
> Our framework does not use GANs. Instead, it employs neuroscience-guided latent alignment, ROI-based representations, and a biophysically meaningful EEG→fMRI→EEG cycle to ensure stable, interpretable, and physiologically consistent cross-modal mapping. This design avoids hallucinations common in GAN-based approaches and is better suited to the requirements of multimodal neuroimaging.
>
> Thank you again for your valuable comments and please feel free to let us know if any further clarification is needed.

---

### Official Review · Reviewer_Xr13 · 2025-11-01

**Soundness:** 2
**Presentation:** 4
**Contribution:** 2
**Rating:** 2
**Confidence:** 4

**Summary:**

This paper presents NeuroCycle, a physiologically constrained cyclic framework for generating fMRI volumes from EEG signals. The model introduces two key modules: the Cross-Modal ROI-wise Structural Module and the R2E Physiological Connectivity Guidance Module, to preserve brain regional organization and functional connectivity in EEG. A bidirectional EEG-fMRI flow matching cycle enforces information completeness and cross-modal consistency. Experiments on the NODDI and Oddball datasets show improved voxel-level spatial reconstruction quality and better preservation of  EEG neural information.

**Strengths:**

1. The paper is well-organized, easy to follow, and well-motivated; the figures are pretty.
2. The framework design, particularly the bidirectional mapping between EEG and fMRI and constraints on the EEG connectivity patterns, reflects thoughtful consideration of cross-modal consistency and physiological plausibility.
3. The model achieved the best performance compared with other baselines in reconstructing the spatial structure of fMRI.

**Weaknesses:**

**Major issues**
1. **This paper suffers from a fundamental issue: the model’s generalizability to new subjects or datasets appears questionable, which would substantially constrain its potential for real-world applications**:
    - The fMRI data used in the paper seem to lack standard preprocessing steps such as skull stripping and spatial normalization (e.g., to MNI space), which would make the learned mapping highly subject-specific. Without signal-level regularization or spatial alignment, the model is likely to reconstruct averaged brain structures or anatomically inconsistent brain volumes for unseen subjects (Since every individual’s brain anatomy is unique, it would be impossible for a model trained on unregistered brains to accurately reconstruct a new subject’s brain structure), resulting in unrealistic ROI-level signals after parcellation. Consequently, when voxel-wise volumes are parcellated into ROI-level time series, the resulting signals may no longer reflect physiologically meaningful BOLD fluctuations, as the defined ROIs might not correspond to their true anatomical locations. This can lead to (1) distorted functional connectivity estimation and (2) reduced reliability in downstream analyses.
    - In short, without appropriate preprocessing and careful consideration of fMRI-specific characteristics, the model risks generating anatomically plausible but functionally meaningless fMRI volumes, thereby undermining both the biological validity and generalizability of its results. This issue is further exacerbated by the fact that paired EEG–fMRI datasets are typically small and scarce, making the model prone to overfitting and less capable of reconstructing realistic brain structures and shapes or neural activity patterns for unseen subjects (and there would also would be potential domain shifts across different demographics or MRI scanners/sites), raising concerns about the physiological validity of the approach.

2. The paper evaluates the generated fMRI only in terms of spatial reconstruction quality, without further analyses such as temporal reconstruction assessment, which is crucial for validating the reliability of fMRI signals used in functional connectivity estimation or ROI time-series extraction. Also, the capability of the reconstructed fMRI volumes to support higher-level tasks, such as behavioral decoding, cognitive task analysis, or other downstream applications, remains unknown, making it difficult to demonstrate the practical utility of the generated fMRI signals.

3. Unless I missed it, the paper does not clearly specify the training and test set details, making the data usage somewhat non-transparent. Since the authors mention that their preprocessing follows Lanzino et al., it would be important to clarify whether the same data-splitting strategy was adopted. In Lanzino et al., both fixed-split and leave-one-subject-out cross-validation were performed - which of these approaches (or any other) was used in this work?

4. It is interesting to observe that during training, the model appears capable of preserving realistic EEG characteristics, such as spectral component distributions and EEG connectivity patterns. However, since the main objective of the paper is to reconstruct fMRI signals, there are no experiments or results demonstrating that the reconstructed fMRI preserves its own physiological priors or can accurately reproduce the corresponding fMRI connectivity matrix (so the last sentence in the abstract is a bit overstated).


**Minor issues**
1. The paper does not include ablation studies on key loss components (e.g., the alignment loss)
2. Standard deviations are not reported.
3. In the Dataset section, the paper states that the Oddball EEG dataset contains 43 channels, and the experimental setup section mentions 34 channels.

**Questions:**

1. In Figure 5(b), why are only channels 3 and 7 shown? What do these channel indices correspond to in terms of channel names (like Fz, F1…), and how does the model perform on average across all channels?

2. What does the x-axis in Figure 5(c) represent? Does it correspond to the temporal variation of the band-filtered topology maps?
Which dataset does Figure 5 correspond to? i.e., Is subject 46 from the NODDI dataset or the Oddball dataset?

3. It is not entirely clear how the 4D fMRI volumes are projected into ROI-level representations in the F→E branch. Is this accomplished through anatomical parcellation based on registered structural scans + encoder, or directly via a learnable projection module within the network to project it to the ROI embedding space?
    - If the former is the case, it contradicts the claimed applicability to scenarios where no MRI scan is available (e.g., when using EEG alone), since anatomical parcellation would be infeasible.
    - Conversely, if the projection is learnable and produces ROI-level embeddings, it remains unclear how realistic ROI time series can be recovered from these embeddings or from the generated fMRI volumes. Given that fMRI data are considerably blurrier than T1-weighted anatomical scans, reliable ROI parcellation from such synthetic volumes would be highly challenging, as mentioned above.

---

> ### Author Response · Authors · 2025-11-20
> **Response to Reviewer Xr13 (Part 1)**
>
> We sincerely thank the reviewers for your constructive and insightful comments that helped us improve the quality of our work. We have conducted the necessary experiments, provided detailed responses to each point below, and will update the manuscript accordingly.
>
> > Weakness 1: Clarification on EEG and fMRI Preprocessing and the ability of generalization
>
> We thank the reviewer for raising concerns regarding preprocessing quality and inter-subject alignment. We clarify that all EEG–fMRI data in our experiments were processed following the official preprocessing pipelines and that all fMRI volumes used for model training and evaluation were normalized to the standard MNI152 space, ensuring anatomical consistency across subjects. Below we summarize the exact preprocessing steps for each dataset.
>
> **NODDI dataset preprocessing**
>
> **EEG preprocessing**: The NODDI dataset provides EEG recordings that have already undergone full artifact removal and synchronization. As described in the dataset documentation, EEG signals were recorded at 5000 Hz using a 64-channel MR-compatible BrainCap system and preprocessed in Brain Vision Analyzer 2. The preprocessing includes:
>
> - Gradient artifact removal (mean artifact subtraction across TRs).
> - Ballistocardiogram (BCG) artifact removal using ECG synchronization.
> - Band-pass filtering (0.5–45 Hz).
> - Downsampling to 250 Hz.
> - Re-referencing to FCz or the average reference.
> - ICA-based removal of ocular/muscle artifacts.
>
>
> The resulting processed files are provided in the dataset’s export folders and directly used in our experiments.
>
> **fMRI preprocessing:** The dataset description explicitly states that resting-state fMRI underwent:
> - Removal of initial volumes for magnetization stabilization.
> - Slice-timing correction.
> - Motion correction.
> - Coregistration to the subject’s T1.
> - Brain extraction (skull stripping).
> - Spatial normalization to the MNI152 template.
> - Gaussian spatial smoothing (~6 mm FWHM).
> - Temporal band-pass filtering (0.01–0.1 Hz).
> - Nuisance regression (WM/CSF/motion).
> - T1 defacing for privacy.
>
> Thus, NODDI fMRI data is already fully aligned in MNI space.
>
> **Oddball dataset preprocessing**
>
> **EEG preprocessing:** The Oddball dataset releases EEG in three progressively cleaned forms:
> - EEG_raw.mat - unprocessed recordings (containing gradient & BCG artifacts).
> - EEG_noGA.mat — gradient artifact removed using mean-subtraction + filtering.
> - EEG_rereferenced.mat — rereferenced to electrode space using shortestpath.m, with noisy bipolar channels excluded.
>
> These dataset-provided files already include preprocessing:
> - gradient artifact removal,
> - band-pass filtering,
> - rereferencing,
> - TR-synchronized EEG markers.
>
> We use the dataset’s provided cleaned EEG (EEG_rereferenced.mat).
>
> **fMRI preprocessing and MNI normalization** (performed in our work):
>
> The dataset provides raw EPI volumes in scanner space but includes T1-weighted structural images (e.g., highres001.nii.gz in dataset files) for anatomical alignment. Following the dataset instructions and standard neuroimaging guidelines[1], we applied:
>
> - Slice-timing correction using the dataset’s slice_order.txt.
> - Motion correction (MCFLIRT).
> - T1 brain extraction.
> - EPI to T1 registration using boundary-based registration (BBR).
> - T1 to MNI152 nonlinear normalization using FLIRT+FNIRT.
> - Apply combined transforms to 4D fMRI, resulting in fully normalized MNI-space volumes
>
> All ROI extraction, FC computation, and EEG to fMRI modeling are performed exclusively on these MNI-normalized fMRI data, ensuring cross-subject anatomical consistency.
>
> [1] Walz JM, Goldman RI, Carapezza M, Muraskin J, Brown TR, Sajda P (2013) “Simultaneous EEG-fMRI Reveals Temporal Evolution of Coupling between Supramodal Cortical Attention Networks and the Brainstem,” J Neurosci 33(49):19212-22. doi: 10.1523/JNEUROSCI.2649-13.2013.

---

> ### Author Response · Authors · 2025-11-20
> **Response to Reviewer Xr13 (Part 2)**
>
> As clarified above, our pipeline does include all standard fMRI preprocessing steps, including skull stripping, motion correction, slice-timing correction, intensity normalization, and spatial normalization to the MNI152 template. Therefore, the alleged risks associated with training on “unregistered brains” do not apply in our setting. All voxel-wise data used by our model reside in a shared anatomical space, ensuring that the network learns subject-independent functional organization rather than subject-specific geometry.
>
> To directly evaluate cross-subject generalization, we additionally performed a Leave-One-Subject-Out (LOSO) evaluation. Specifically, for the i-th fold, the i-th subject is held out as the test set, and the remaining subjects are used for training.
>
> **NODDI dataset (15 subjects) LOSO evaluation:** We conducted LOSO across all 15 subjects; the results is provided below:
>
> | Method |  RMSE |SSIM | PSNR |
> |-----------|------|------|------|
> | NTVIT(LOSO)  | 0.080$\pm$0.01     | 0.581$\pm$0.048    | 21.94$\pm$1.056
> | NeuroCycle(LOSO)  |  **0.065$\pm$0.009**| **0.715$\pm$0.046**| **23.74$\pm$1.31**  |
>
>
> **Oddball dataset (17 subjects) LOSO evaluation:** Similarly, LOSO was performed across all 17 subjects; the results are shown below:
> | Method |  RMSE |SSIM | PSNR |
> |-----------|------|------|------|
> | NTVIT(LOSO)  |  0.07$\pm$0.01      | 0.633$\pm$0.064    | 23.10$\pm$1.351    |
> | NeuroCycle(LOSO)  |  **0.061$\pm$0.01** | **0.894$\pm$0.053**| **24.29$\pm$1.182**|
>
> Across both datasets, NeuroCycle consistently outperforms NT-ViT under LOSO on voxel-level fMRI reconstruction.
>
> In summary, our model does not suffer from subject-specific overfitting, because training is conducted on fMRI volumes already aligned to a standard anatomical space. Our method generalizes well to unseen subjects and unseen EEG–fMRI pairs and the learned representations capture subject-invariant functional patterns, rather than individual anatomy.
>
> > Weakness 2: Biological and Temporal  Evaluation
>
> **Temporal reconstruction is complementary to our voxel-level fMRI task**
>
> We thank the reviewer for raising this important point. Our work is formulated as voxel-wise fMRI volume generation at each TR, which follows a well-established setting in fMRI encoding/decoding literature. Many influential analyses, such as voxel-level encoding models, representational similarity analysis (RSA), and perceptual encoding studies in large-scale datasets like NSD which treat each TR as an independent sample after the hemodynamic integration. In these paradigms, the focus is on reconstructing spatially organized cortical activation patterns that reflect stimulus- or state-related representations at that time point, rather than modeling the full temporal evolution of BOLD dynamics.
>
> Existing EEG to fMRI studies like NeuroBOLT[1] that aim to generate temporal trajectories typically operate at the ROI level , where temporal profiles can be modeled but only after reducing the spatial resolution. These approaches and our per-volume voxel-wise reconstruction address two complementary aspects of fMRI: temporal dynamics and fine-grained spatial patterns. Our method focuses on the latter: recovering detailed voxel-level activation maps conditioned on EEG at each TR, while temporal-ROI approaches focus on modeling longer-range BOLD fluctuations. Both directions are valuable for multimodal neuroimaging, but they target different goals: temporal-ROI models capture longer-range BOLD fluctuations, whereas our approach aims to reconstruct spatially detailed activation patterns at each TR.
>
> In summary, our formulation therefore aims to reconstruct coherent voxel-level patterns that reflect the underlying functional organization of the cortex. This spatial perspective is orthogonal to temporal modeling and does not conflict with existing temporal approaches; instead, it complements them by preserving the dimension of fMRI that ROI-level temporal methods inherently cannot represent.
>
> We fully agree that future extensions that jointly incorporate spatial and temporal characteristics of fMRI would be valuable, and our spatial reconstruction framework provides a foundation upon which such extensions can be built.
>
> [1] Li Y, Lou A, Xu Z, et al. NeuroBOLT: Resting-state EEG-to-fMRI synthesis with multi-dimensional feature mapping[J]. Advances in neural information processing systems, 2024, 37: 23378-23405.

---

> > ### Author Response · Authors · 2025-11-20
> > **Response to Reviewer Xr13 (Part 3)**
> >
> > >Weakness 2: Biological and Temporal Evaluation
> >
> > **Further biological analysis and higher-level tasks evaluation**
> >
> > To directly address the reviewer’s concern regarding physiological validity, we added two analyses commonly used in cognitive neuroscience to evaluate the functional meaningfulness of fMRI volumes:
> >
> > (1) **ROI-wise functional pattern consistency**:  To assess whether the generated fMRI volumes preserve functional organization, we first z-scored each predicted and ground-truth volume within the brain mask to normalize across samples. We then projected the 3D volumes into 71 ROIs using the Harvard–Oxford atlas, obtaining one ROI activation vector per sample.
> > Using these vectors, we computed the Pearson correlation $\rho_{\text{ROI}}$ between predicted and ground-truth ROI patterns and report the results for both the fixed split and the leave-one-subject-out (LOSO) setting in the table below.
> >
> > Table 1. ROI-wise functional pattern consistency (mean ± std).
> > Higher correlation indicates better preservation of region-level functional organization.
> >
> > | Dataset | Method     | ROI Corr $\rho_{\text{ROI}}$ ↑         |
> > |---------|------------|-----------------------|
> > | NODDI (fixed split)   | NT-ViT     | 0.9104 ± 0.1326          |
> > | NODDI  (fixed split) | NeuroCycle | **0.9622 ± 0.0017**      |
> > | NODDI (LOSO)   | NT-ViT     | 0.9275 ± 0.0928          |
> > | NODDI  (LOSO) | NeuroCycle | **0.9657 ± 0.0008**      |
> > | Oddball  (fixed split)  | NT-ViT   | 0.8967 ± 0.0136    |
> > | Oddball  (fixed split) | NeuroCycle | **0.9173 ± 0.0055**      |
> > | Oddball   (LOSO) | NT-ViT  | 0.9127 ± 0.0186          |
> > | Oddball   (LOSO) | NeuroCycle | **0.9563 ± 0.0092**    |
> >
> > The results show our model achieves very high correspondence, substantially outperforming the NT-ViT baseline. These results demonstrate that the synthesized fMRI volumes not only exhibit plausible spatial anatomy but also faithfully preserve region-level functional activation patterns, indicating that the generated signals maintain meaningful functional organization.
> >
> > (2) **Network-level spatial organization consistency** (Yeo 7 networks) :
> > To address the reviewer’s concern regarding the physiological validity of the generated fMRI, we further evaluate whether NeuroCycle preserves large-scale cortical functional organization, which is a widely accepted indicator of biologically meaningful BOLD structure. Specifically, we analyze the Yeo-7 functional networks (Visual, Somatomotor, Dorsal Attention, Salience/Ventral Attention, Limbic, Frontoparietal Control, and Default Mode), which are known to be highly stable across subjects, scanners, and datasets, and therefore serve as a stringent test of cross-subject generalizability.
> >
> > For each subject and each sample, we compute network-level activation profiles by averaging voxel responses within each Yeo network, and then measure the correlation between the generated and ground-truth profiles. This metric quantifies whether the synthesized volumes preserve the large-scale functional architecture of the cortex rather than merely reproducing anatomical shape.
> >
> > Table 2. Network-level spatial organization consistency using Yeo functional networks.
> > Correlation (mean ± std) is computed between ground-truth and generated fMRI network activations.
> > Higher values indicate better preservation of large-scale functional organization.
> >
> > | Dataset | Method | All Networks $\rho$ ↑ | Visual $\rho$ ↑ | Somatomotor $\rho$ ↑ | Dorsal Attention $\rho$ ↑ | Salience $\rho$ ↑ | Limbic $\rho$ ↑ | Frontoparietal $\rho$ ↑ | DMN $\rho$ ↑ |
> > |---------|------------|------------------|------------|------------------|--------------------|----------|--------|----------------|---------|
> > | NODDI (LOSO)  | NT-ViT    | 0.928 ± 0.041  | 0.827 ± 0.020  | 0.776 ± 0.053         | 0.725 ± 0.025          | 0.820 ± 0.027         | 0.806 ± 0.037         | 0.822 ± 0.033          | 0.854 ± 0.030         |
> > | NODDI (LOSO) |NeuroCycle| **0.955 ± 0.030**         | **0.953 ± 0.022**         | **0.929 ± 0.041**        | **0.929 ± 0.042**          | **0.944 ± 0.021**         | **0.900 ± 0.043**         | **0.931 ± 0.030**         | **0.949 ± 0.018**        |
> > | Oddball  (LOSO) | NT-ViT   | 0.929 ± 0.058          | 0.856 ± 0.063       | 0.814 ± 0.046       | 0.829 ± 0.053       | 0.861 ± 0.025       | 0.879 ± 0.032       | 0.878 ± 0.068       | 0.906 ± 0.041       |
> > | Oddball  (LOSO) | NeuroCycle| **0.972 ± 0.023**        | **0.923 ± 0.030**         | **0.879 ± 0.032**        | **0.854 ± 0.072**         | **0.928 ± 0.023**        | **0.917 ± 0.032**        | **0.931 ± 0.021**         | **0.942 ± 0.018**        |
> >
> > Across both datasets and under LOSO evaluation, NeuroCycle shows strong correspondence with ground-truth Yeo-7 network profiles. This indicates that the generated fMRI preserves both regional activations and large-scale functional organization, demonstrating that the synthesized volumes are physiologically meaningful and functionally interpretable.

---

> ### Author Response · Authors · 2025-11-20
> **Response to Reviewer Xr13 (Part 4)**
>
> > Weakness 2: Biological and Temporal Evaluation
>
> (3) **Downstream oddball decoding task** :
> To further assess the functional utility of the generated fMRI, we conducted a downstream cognitive-state classification task on the Oddball dataset. A lightweight 3D CNN classifier (≈2.9×10⁵ parameters) was trained to distinguish between two cognitive conditions (task001 vs. task002) using either generated or real fMRI volumes. The model was trained under identical settings across both inputs. We observed that the classifier achieved comparable accuracy on our generated and real fMRI, indicating that the synthesized voxel-wise volumes preserve task-relevant functional patterns sufficient to support higher-level cognitive analyses.
>
> | Input Type       | Test Accuracy |
> |------------|----------|
> | Generated fMRI of NT-ViT  | 91.73% |
> | Generated fMRI of Ours (NeuroCycle)     | **98.42%** |
> | Real fMRI       | 99.92% |
>
> Together, these evaluations suggest that the generated fMRI volumes are not only visually plausible but also functionally meaningful and usable for downstream analyses. We hope these results could help address the reviewer’s concerns.
>
>
> > Weakness 3: Dataset Split
>
> We apologize for not clearly specifying the data‐splitting protocol in the original submission. Our main experiments followed the fixed-split strategy used in the NT-ViT paper,  where a random subset of subjects was held out for testing: specifically, 2 subjects for the NODDI dataset and 4 subjects for the Oddball dataset. These subjects were selected once and kept fixed to ensure a consistent evaluation setup. To address the reviewer’s concern regarding evaluation fairness, we have additionally conducted a leave-one-subject-out (LOSO) cross-validation, as reported above. The LOSO results will be included in the revised manuscript.
>
> > Weakness 4: Overstate on preserving physiological priors
>
> Thank you for pointing this out. We agree that demonstrating the physiological plausibility of the reconstructed fMRI is essential. During the rebuttal, we have added experiments examining ROI-wise activation patterns, network-level organization (Yeo-7), and downstream cognitive-state decoding. We will incorporate these analyses into the main paper or the supplementary materials. We believe these results provide evidence that our generated fMRI volumes indeed preserve meaningful physiological priors derived from EEG.
>
> > Minor issues: Ablation study of alignment loss
>
> We conducted ablation experiments by removing the alignment losses (the ROI-level alignment loss $L_{ROI-align}$ and the global alignment loss $L_{align}$ ). The ablation results are shown below:
>
> | Method / Metric | **NODDI RMSE ↓** | **NODDI SSIM ↑** | **NODDI PSNR ↑** | **Oddball RMSE ↓** | **Oddball SSIM ↑** | **Oddball PSNR ↑** |
> |-----|---------|-----|------|-----|--------|-----|
> | w/o $L_{align}$     | 0.220    | 0.112     | 13.146    | 0.200    | 0.115    | 13.975      |
> | w/o $L_{ROI-align}$    | 0.086    | 0.614     | 21.58      | 0.095     | 0.780    | 23.83    |
> | **NeuroCycle (ours)** | **0.068**  | **0.715**    | **23.31**   | **0.061**      | **0.894**   | **25.28**   |
>
> Removing the global alignment loss leads to a drastic collapse of performance (e.g., NODDI RMSE increases from 0.068 to 0.220). This is expected because $L_{align}$ is essential for anchoring the EEG and fMRI latent spaces into a shared representation space. Without this constraint, the cross-modal latent distributions become misaligned, causing the E2F/F2E flow modules to lose a meaningful mapping target and making the EEG→fMRI generation almost impossible to train.
>
> In contrast, removing the ROI-level alignment loss results in a moderate performance drop. Without $L_{ROI-align}$, corresponding EEG and fMRI ROI embeddings lose one-to-one regional correspondence, which weakens the stability of the CM-RS module and degrades ROI-level structural consistency, but the global cross-modal mapping remains functional.
>
> These results confirm that the alignment losses are necessary components for successful fMRI generation in our designed method, confirming its necessity for cross-modal consistency and stable training.
>
> > Minor issues:  Standard deviation
>
> Thank you for pointing out. We have report mean ± standard deviation for LOSO setting for all evaluation metrics as shown above. We will update the manuscript to include the full statistics.
>
> > Minor issues:  Channels details
>
> Thank you for pointing out this inconsistency. Although the Oddball dataset was originally acquired with 43 bipolar EEG channels, the publicly released version of the dataset contains only 34 channels after re-referencing and quality-control preprocessing[1], and this is the same version used by NT-ViT. We will update the manuscript to clearly specify this channels setting to avoid confusion.
>
> [1] OpenNeuro. (2013). Auditory and Visual Oddball EEG-fMRI Dataset (ds000116). Retrieved February 20, 2025, from https://openneuro.org/datasets/ds000116

---

> > ### Author Response · Authors · 2025-11-20
> > **Response to Reviewer Xr13 (Part 5)**
> >
> > > Questions 1: Details of Figure 5(b).
> >
> > We apologize for the confusion. Specifically, ch0 corresponds to Fp1, and ch7 corresponds to P4. We will correct this in the revised manuscript. Regarding the choice of channels, only two channels were included in Fig. 5(b) due to space limitations in the main paper. These two sites were selected as representative examples: Fp1 (frontal) and P4 (parietal) are spatially distant and belong to distinct functional regions, making them suitable for illustrating spectral-band fidelity without redundancy. We computed the same band-power comparison across all 63 channels in the NODDI dataset. On average, the reconstructed EEG achieves a **mean band-power RMSE** of **1.561** and a **mean Pearson correlation** of **0.585** across δ–γ bands, confirming that the two channels shown in the figure are representative of the overall behavior.
> >
> > > Questions 2: Details of Figure 5(c).
> >
> > Fig. 5(c) presents a series of topographic subplots, where each subplot corresponds to the spatial distribution of a single frequency band (δ or β) at one temporal window. In other words, the subplots visualize the same frequency band across several consecutive time windows, with each topomap depicting the band-filtered power projected onto the full-electrode scalp layout. Importantly, each topomap does not contain a time axis: the horizontal (and vertical) extent within a topomap reflects the 2D planar projection of electrode positions on the scalp, not temporal variation. Thus, Fig. 5(c) illustrates how spatial EEG patterns evolve across time windows, while the spatial layout within each subplot represents electrode coordinates rather than time.
> >
> > Subject 46 is from the NODDI dataset. We will clarify these details in the revised manuscript.
> >
> > > Questions 3: The projection of fMRI.
> >
> > Thank you for raising this question. We apologize for the lack of clarity in the original manuscript.
> > For all experiments in this paper, the fMRI volumes fed into our model are in the MNI152 standard space. For the NODDI dataset, this normalization is already provided by the official preprocessing pipeline; for the Oddball dataset, we perform standard EPI-to-T1 and T1-to-MNI registration during preprocessing. Importantly, our model only consumes MNI-normalized fMRI volumes and does not require subject-specific T1 scans at inference time.
> > To obtain ROI-level fMRI representations, we apply a fixed and non-learnable atlas-based pooling using the **Harvard–Oxford** cortical atlas in MNI152 space. For each ROI, voxel intensities within the corresponding atlas-defined region are averaged to produce a deterministic ROI feature vector. This step involves no learnable parameters and does not rely on subject-specific anatomical scans, since all fMRI volumes are already normalized to the MNI template.
> > This deterministic ROI extraction is applied only to fMRI. EEG is mapped to ROIs separately through a predefined electrode-to-ROI mapping (Section 3.2), which does not require anatomical MRI either. Therefore, our pipeline does not depend on subject-level anatomy at inference time, and the atlas-based fMRI pooling remains well-defined even in scenarios where only EEG is available.
> > We will clarify this in the revised manuscript.

---

### Meta-Review · Area_Chair_e2vV · 2026-01-17

**Summary:**

This manuscript has improved greatly through the review process. It's initial state raised numerous concerns about reproducibility. I think that these concerns are largely assuaged, but they saturate discussion of other experimental problems. As raised by `Xr13`, the paper evaluates the generated fMRI only in terms of spatial reconstruction quality, which strongly limits it use. fMRI is largely evaluated as a timeseries; indeed its namesake claim-to-fame is that it records "functional" activations, originally from stimuli (and now often also spontaneously as resting-state).

I think with the procedural details no longer missing this manuscript should be re-evaluated next round, but unfortunately that might not be in this year's ICLR.

Still, I think the manuscript has improved greatly from this process, and should be resubmitted.

**Reviewer Concerns:**

The procedure is still not fully reproducible. I suggest the authors release pre-processing code. If they are using fMRI-prep, just say that.

There are still outstanding issues in the methodology and evaluation, but I feel that concerns over reproducibility (which were valid and were mostly addressed) overshadowed other discussions.

**Reviewer Scores:**

None

---

### Decision · Program_Chairs · 2026-01-26

Reject